# Robust mapping of spatiotemporal trajectories and cell–cell interactions in healthy and diseased tissues

Duy Pham[1], Xiao Tan[1], Brad Balderson[1,2], Jun Xu[3], Laura F. Grice[1,4], Sohye Yoon[3], Emily F. Willis[4], Minh Tran[1], Pui Yeng Lam[1], Arti Raghubar[1], Priyakshi Kalita-de Croft[5], Sunil Lakhani[5], Jana Vukovic[4,6], Marc J. Ruitenberg[4] ✉ & Quan H. Nguyen[1,7] ✉

Spatial transcriptomics (ST) technologies generate multiple data types from biological samples, namely gene expression, physical distance between data points, and/or tissue morphology. Here we developed three computational-statistical algorithms that integrate all three data types to advance understanding of cellular processes. First, we present a spatial graph-based method, pseudo-time-space (PSTS), to model and uncover relationships between transcriptional states of cells across tissues undergoing dynamic change (e.g. neurodevelopment, brain injury and/or microglia activation, and cancer progression). We further developed a spatially-constrained two-level permutation (SCTP) test to study cell-cell interaction, finding highly interactive tissue regions across thousands of ligand-receptor pairs with markedly reduced false discovery rates. Finally, we present a spatial graph-based imputation method with neural network (stSME), to correct for technical noise/dropout and increase ST data coverage. Together, the algorithms that we developed, implemented in the comprehensive and fast stLearn software, allow for robust interrogation of biological processes within healthy and diseased tissues.

Biological tissues represent enormously complex and dynamic cellular ecosystems, the functions of which are driven by cell type(s), their local composition and states, distribution patterns, and cell–cell interactions[1–3]. The nature of these features at any given place and time are critical determinants of tissue development, homeostasis, repair and responses to environmental signalling[1,4]. The advent of single-cell RNA sequencing (scRNA-seq), an ultra-sensitive and high-throughput technology with individual cell resolution[5], has led to the discovery of new cell types and also expanded our understanding as to how the transcriptional state(s) of cells can vary in response to experimental stimuli and/or changes in their environment. However, current knowledge about cell types and states often still lacks crucial contextual information, that is, how they coexist, interact and communicate within their native tissue environments in either healthy or diseased states[6–8].

Spatial transcriptomics (ST) can profile transcriptome-wide gene expression in an unbiased manner without the need for tissue dissociation, thus retaining spatial information. ST data is growing exponentially[9,10], with the technology now becoming more widely accessible through platforms such as Visium[11], NanoString Spatial Profiling[12], seqFISH+[13], MERFISH[14] and Slide-seq2[15]. However, analytical methods to analyse such complex datasets have lagged behind experimental advances and mostly remain in an early development

[1]Institute for Molecular Bioscience, The University of Queensland, Brisbane, Australia. [2]School of Chemistry and Molecular Biosciences, The University of Queensland, Brisbane, Australia. [3]Genome Innovation Hub, The University of Queensland, Brisbane, Australia. [4]School of Biomedical Sciences, Faculty of Medicine, The University of Queensland, Brisbane, Australia. [5]UQ Centre for Clinical Research, The University of Queensland, Brisbane, Australia. [6]Queensland Brain Institute, The University of Queensland, Brisbane, Australia. [7]QIMR Berghofer Medical Research Institute, Herston, Australia. ✉e-mail: m.ruitenberg@uq.edu.au; quan.nguyen@uq.edu.au

stage. For example, morphology and gene expression are known to be strongly linked[16], and our own previous work indeed demonstrated that the use of imaging or gene expression data alone is less accurate at predicting cell types and/or disease stage compared to models that combine both data types[17]. Most existing analysis methods for ST data, however, still do not combine spatial and imaging information with gene expression data to jointly study important processes like cell–cell communication and/or spatial changes in cell states (trajectories). Finding patterns in spatial gene expression data thus remains one of the grand challenges in omics data science today[8,18].

In this study, we developed a powerful and flexible approach to integrate gene expression measurements with the spatial location and/or morphological information, to effectively make use of all dimensions in ST data. The analytical toolkit that we describe, hereafter collectively referred to as stLearn, addresses three major research questions around understanding biological processes within tissue sections: (1) the (re)construction of spatio-temporal trajectories, (2) the study of cell–cell interactions, and (3) the improvement of spatial data quality by imputation. We show that the inclusion of spatial information and morphological data can address current challenges in each of these three research areas with higher accuracy than existing methods and/or add analysis capabilities that are not yet available (e.g. cancer progression trajectory analysis). stLearn can be used with most spatial data, even those that lack tissue image information. stLearn's assumptions are based on existing biological knowledge and principles. Specifically, the interdependence between gene expression and morphological features such as cell size, nuclei size, granularity and distribution[16,19] is used to adjust gene expression values. Physical distance is used on the basis that genes in cells that are nearby within a given tissue display more similar expression patterns than distant cells[20–22]. Regions of increased cell type diversity also correlate with higher cell–cell interaction activities, as demonstrated for example by the immune social network model, or the weighted-directed-multi-hyperedge network model[23,24]. By incorporating all this information, our approach delivers significant improvements over existing methods in multiple criteria, for example by providing the capability to find spatial trajectories within (and across) tissues, and also by allowing a critical reduction in the detection of false positive cell–cell interactions within ST data.

## Results

### An interpretable graph-based framework to contextualise gene expression data with spatial neighbourhood and/or morphological information

stLearn implements a graph-based framework to flexibly integrate two or all types of information available in ST data, that is, gene expression, tissue morphology (optional) and physical distance (Fig. 1a); this graph-based framework is interpretable as the individual contribution of each type of information can be quantified. stLearn can analyse a wide range of spatial transcriptomics data types, with or without imaging information (Fig. 1b), and its three main algorithms allow users to infer spatial trajectories that recapitulate changes in biological processes connecting neighbouring cells across the tissue (Fig. 1c and S1), to map significant spatial cell–cell interactions (Fig. 1d), and to impute spatial gene expression data (Fig. 1e). The biological applications for each of these three algorithms are demonstrably broad, and we thoroughly tested and validated our methods in a wide range of biological systems using in-house, public and simulated datasets, as described below.

### stLearn reconstructs spatio-temporal cell trajectories in brain injury, neurodevelopment and cancer

Our spatial trajectory inference algorithm, pseudo-time-space (PSTS), allows users to deduce changes in cell state across tissue space and time (Fig. S1 and Supplementary Note 1). One drawback of scRNA-seq data is that anatomical information about a cell's location within the broader tissue is lost, as is context from the local cellular neighbourhood. Furthermore, trajectory reconstructions in scRNA-seq data are generated under the assumption that all cells of the same cell type developed from similar progenitor and/or cell states. However, this assumption does not hold if one cell type is in fact distributed across different regions, or where region-specific changes for that cell type may occur; examples of this would include instances of tissue injury and inflammation as well as metastatic tumours. This shortcoming can be resolved with ST, if gene expression information is coupled to cellular distribution data (Fig. S1, and also discussed later in Figs. 2 and 3, and S10). We therefore created the PSTS algorithm to reconstruct spatial trajectories that can track pseudo-temporal patterns across a tissue in ST datasets (Supplementary Note 1).

We hypothesised that our PSTS trajectory algorithm would be able to detect (and predict) spatio-temporal responses to tissue injury, specifically gradients of microglia activation in a well-characterised mouse model of traumatic brain injury (TBI)[25]. Under steady-state conditions, these resident macrophage-like cells of the brain display little heterogeneity between different brain regions and have a mostly ramified phenotype[26,27]. They rapidly change their gene expression and morphology, however, in response to insult, becoming visibly more amoeboid in appearance (Fig. 2a). We therefore further hypothesised that we would be able to validate PSTS predictions morphologically, using microglia density and size as a proxy for their activation.

To test these hypotheses, we first generated Visium ST data for the injured mouse brain (3 days post-injury, dpi). We then applied stSME-based clustering (see "Methods") to segment the brain (Figs. S2 and 2c), using the Allen Mouse Brain Atlas for the fine-tuning of clustering parameters[28], and subsequently selected all microglia-containing spots based on the expression of marker genes *Fcrls* and *Tmem119*. When applying PSTS to these spots, the hypothalamus region was revealed to be the most transcriptionally dissimilar to the injury site, as captured by its $d_{PTS}$ score (Fig. S3). We then used the PSTS algorithm to predict the minimum spanning tree connecting the damaged site and the hypothalamus (refer to Methods and Supplementary Note 1.3). This yielded a spatial trajectory for microglial activation across the dorsoventral axis of the injured brain, with the arrows indicating the directionality of transcriptional change in PSTS values (Fig. 2b). Based on matching clusters with the anatomical identity of brain regions (Fig. 2c), key nodes within the PSTS trajectory for microglia activation were the hypothalamus (node 4), thalamus (node 2), hippocampus (node 3) and two branches to the penumbra region on either side of the lesion (nodes 1) (Fig. 2b); path-defining genes are shown in Fig. 2d. Enrichment analysis revealed the microglia pathogen phagocytosis pathway as the most significant biological process changing across our spatial trajectory (Fig. 2e). Other relevant pathways, including those involved in the TYROBP causal network, oxidative stress and central nervous system (CNS) injury more broadly, also changed with microglial activation.

Detailed histological studies of microglia morphology and density across six different conditions and/or time points (sham control, 6 hours, 1 day, 3 days, 5 days and 12 dpi) independently validated our 3 dpi PSTS trajectory, with changes in microglia number, cell body size and shape matching the prediction across both space and time (Fig. 2f-h). We corroborated this further with additional ST data by also mapping the expression of gene markers associated with microglia activation at two time points post-TBI, i.e. 6 hours (Legacy ST, with lower resolution) and 3 days (Visium and Legacy ST platforms), as well as in a non-injured ST brain sample (Visium ST) for control purposes (see Fig. S4 and also Fig. S6c). A clear injury-induced shift in the spatial expression of microglia markers *Fcgr*1, *C1qa* and *Cyba* could be observed between the control, 6 hours and 3 days post-TBI samples, with expression increasing over time closer to the damaged site (Fig. S4).

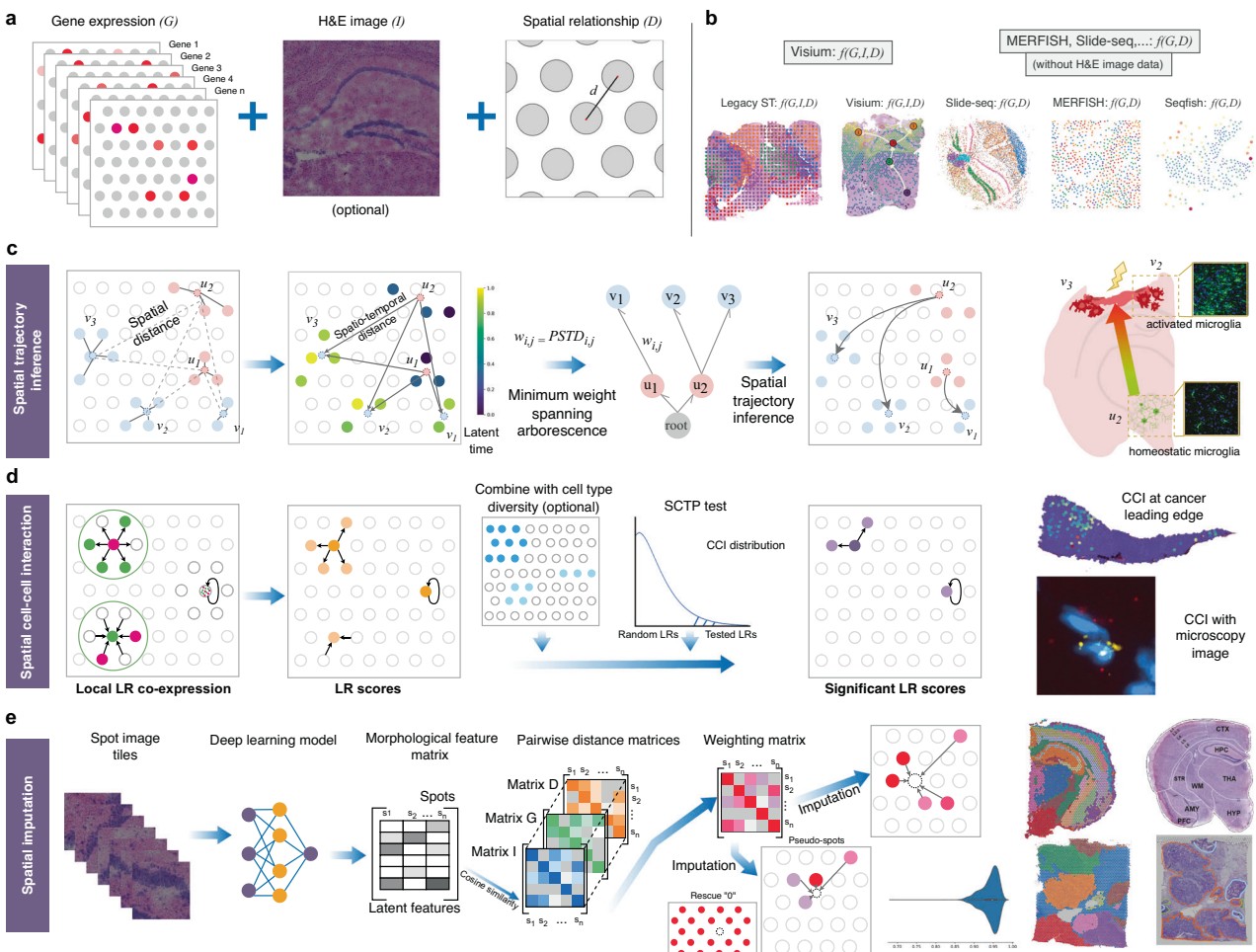

**Fig. 1 | Spatial analysis algorithms implemented in stLearn. a** Schematic diagram showing the three spatial data types that can be integrated by stLearn: gene expression (G), imaging (I) and spatial distance (D). **b** stLearn can be applied to a range of spatial technologies, with or without tissue imaging information (using $f(G,I,D)$ or $f(G,D)$ functions). **c** Spatial trajectory analysis to infer biological processes within an undissociated tissue. Pseudo-space-time distance (PSTD) values are calculated based on gene expression and physical distance. Spatial distance is calculated between the centroid coordinates of clusters $U$ and $V$ with sub-clusters $(u_1, u_2)$ and $(v_1, v_2, v_3)$. PSTD values are used to construct a rooted, directed graph (arborescence), the topology of which can be optimised by a minimum spanning tree to infer the trajectory. This approach to trajectory analysis was validated in a mouse model of traumatic brain injury. **d** Spatially-constrained two-level permutation (SCTP) analysis for cell–cell interaction (CCI) between (straight arrows) and within (looped arrows) spatial spots. SCTP uses ligand and receptor co-expression

information among neighbouring spots, and cell type diversity (gradient blue spots; darker colour indicates more cell types per spot) to compute ligand-receptor (LR) scores. SCTP finds hotspots (purple) within a given tissue, where LR interactions between cell types are more likely to occur compared to a null distribution of random non-interacting gene-gene pairs. Predicted interactions were confirmed by RNA single molecule imaging. **e** Overview of within-tissue imputation and clustering by stSME, which corrects for technical noise (dropouts) in gene expression values by using imaging data (via a neural network model - matrix I), and spots that are both physically near and have similar gene expression profiles (distance matrices D and G, respectively). stSME can also predict gene expression in tissue regions for which there is no experimental data (pseudo-spots). stSME clustering performance was validated against an established anatomical reference mouse brain (spatial brain data, top far right), or expert pathologist annotation (breast cancer data, bottom far right).

For benchmarking, we next compared PSTS/pseudotime results between tools, visualising the variation of PSTS/pseudotime values between spatial spots within the tissue based on variogram metrics see "Methods" and Supplementary Note 1.2 for details). We find that PSTS outperformed Slingshot[29] and Monocle3[30], which are non-spatial trajectory inference methods (Fig. 2i). Specifically, PSTS constructed more meaningful trajectories compared to Slingshot and Monocle3 (the method used by SPATA[31]) (Fig. 2j); we validated this through cell type annotation by deconvolution and experimental histological studies (Fig. S5). To also benchmark against other pseudotime methods that do use spatial information, we next compared the performance of PSTS to that of SpaceFlow[32] (Figs. 2j and S6). While SpaceFlow's pseudo-Spatiotemporal Map (pSM) did provide spatially smooth gradients with less variation between neighbouring spots (indicated by a low semivariance in the variogram; Fig. S6a), the smoothed pSM scores across spots did not reveal the gradient of microglia activation relative

to the damage site (Fig. 2j and S4–5). SpaceFlow's spatial regularisation and/or a potential loss of information on spot-to-spot variation in the latent space after dimensionality reduction (which is used by Space-Flow to calculate pseudotime scores) may have contributed to this issue. In our analysis, the pseudotime values computed by SpaceFlow did not form a pattern that enabled the drawing of a tree for optimising the trajectory from low to high pseudotime scores, a unique feature of stLearn. The SpaceFlow result also did not allow us to identify biologically significant transition genes along the trajectory, and the pathways associated with these (Fig. S6d). Overall, PSTS thus outperformed all other trajectory inference methods tested, including the method that uses spatial information (Fig. 2j).

We next assessed PSTS' ability to also reveal spatio-temporal trajectories under normal (non-injury) conditions using mouse embryonic brain development as the model. For this, we applied PSTS to an existing mouse embryonic day 14 sci-Space dataset[33] (Figs. 3a, b

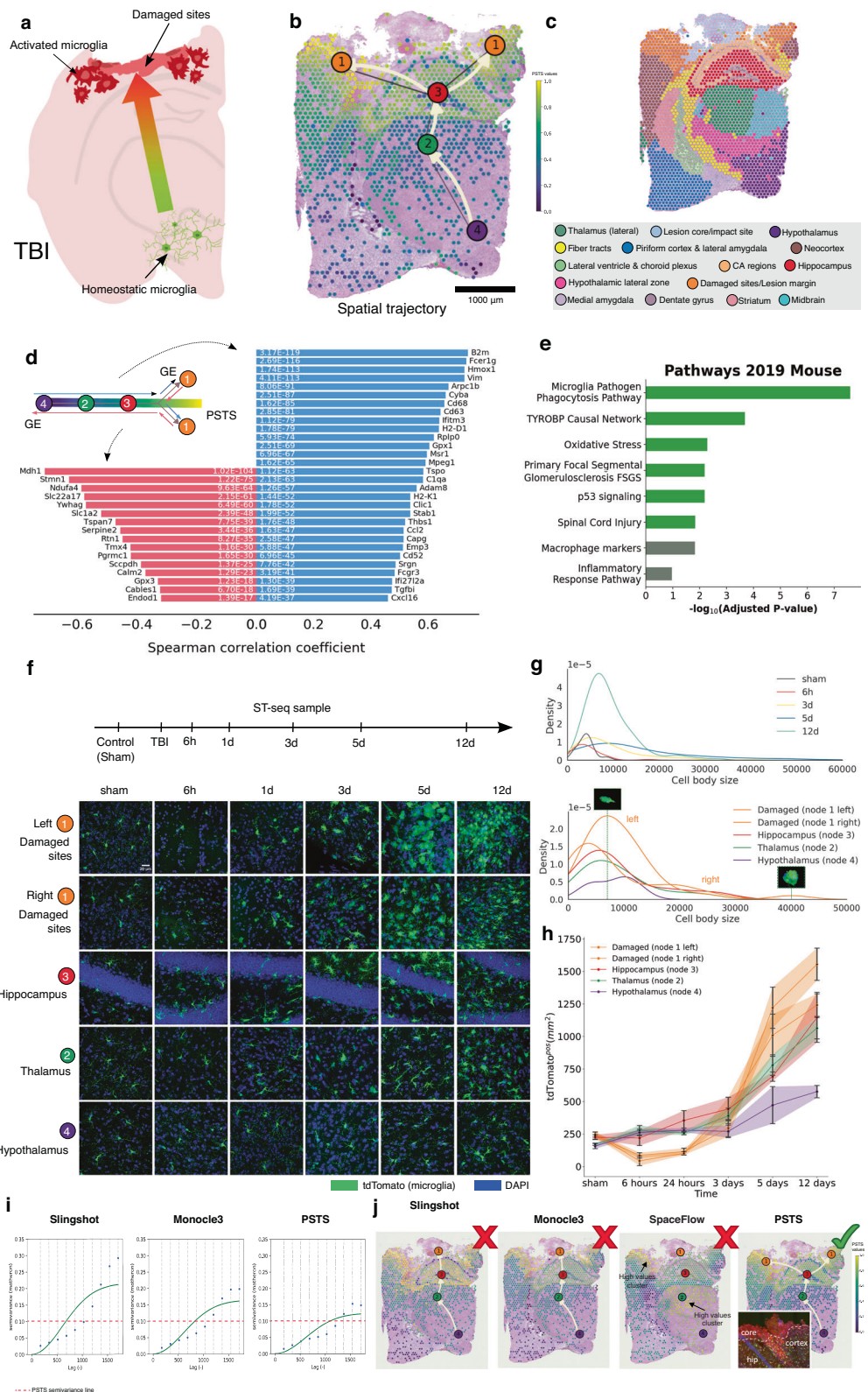

and S7 and Supplementary Note 1.6). Previous analysis of brain development in this dataset identified trajectories (without spatial information), running from radial glia to neurons[33]. Our newly calculated PSTS values and trajectories were consistent with these previously reported results, with the spatio-temporal gradients reflecting the well-documented pattern of cellular differentiation, neuronal migration and maturity. Importantly, PSTS found one additional branching pattern, which precisely illustrated immature neurons

migrating and differentiating radially outward as part of the inside-out development of the cortical layers (Figs. 3b and S7).

We lastly assessed PSTS' utility in a more applied, diagnostic context, namely its ability to model the metastasis potential of ductal carcinoma in situ (DCIS) cells in breast cancer (detailed in Supplementary Note 1.5). Here, PSTS revealed the relationship between ductal and invasive states in breast cancer. DCIS are abnormal cells in the breast duct that have the potential to metastasise beyond the duct, becoming

**Fig. 2 | Pseudo-time-space (PSTS) trajectory analysis and validation in a mouse model of traumatic brain injury (TBI). a** Schematic showing the cortical impact site and microglia activation. **b** Spatio-temporal trajectory of microglial activation at 3 days post-TBI, as predicted by our PSTS algorithm, running from the hypothalamus (node 4), through the thalamus (node 2) and hippocampus (node 3) and then the cortical penumbra regions adjacent to the lesion core (nodes 1). Colour-coded pseudo-time-space values (ranging from *0* to *1*) reflect microglia-related gene expression changes through the tissue space. **c** Clustering results for TBI Visium ST data (*n* = 2442 spots). **d** Transition genes positively (blue) or negatively (red) correlated with the predicted trajectory for microglia activation (extracted by Spearman correlation test of pseudo-time-space values; adjusted *p*-value < 0.05 and correlation coefficient >0.3 or <−0.3). **e** Enrichment analysis of upregulated transition genes revealing significant pathways related to microglia activation, inflammation and neural injury. **f** Experimental validation of the spatio-temporal trajectory for microglia (green) activation following TBI; cell nuclei are shown in blue. Imaging was performed across five different brain regions of interest (ROIs;

from one brain per time point), equivalent to the trajectory nodes, from sham (uninjured) controls and five different time points post-TBI. Note the changes in microglia abundance and morphology across cluster nodes and time. **g** Density plots illustrating changes in microglia cell body size (proxy for activation) over time (top) and space (bottom; 3 days post-TBI only). **h** Changes in microglia density over time and space for all ROIs (*n* = 4 biological replicates per time point; error bars show SEM. **i** Variograms depicting the autocorrelation of PSTS/pseudotime values for each spot. Plots show the spatial variance in PSTS/pseudotime values produced by Slingshot, Monocle 3 and PSTS. Lower values of the semi-variance Matheron estimator indicate higher PSTS/pseudotime continuity in the spatial context, and thus a more likely trajectory (see "Methods"); PSTS semi-variance is indicated by the red dashed line. **j** Spatial branching patterns for microglia activation using different trajectory analysis methods. Only PSTS predicted a trajectory leading to the penumbra regions rather than the core (where microglia are mostly absent; see inset and also Figs. S5 and S6).

---

invasive ductal carcinomas (IDCs). Being able to visualise and/or predict DCIS-to-IDC progression in space and time has the potential to discover druggable biological pathways and/or biomarkers of disease, and with that, improvements in clinical care. We find that PSTS was able to find different states of cancer cells present within a given breast cancer tissue section (Figs. S8 and S9), and that it could model the potential transition between these states (Figs. 3c–f and S10–S11). By inferring branches of spatial trajectories, PSTS suggested the most likely progressions of ductal states (clusters) to spatially corresponding invasive states, as well as the distinct pathways and genes associated with these branches (Figs. 3c–e, S11, S12). To also examine here if spatial trajectories translate and/or can be inferred across different tissues, we devised a broadly applicable integration strategy. This integration strategy harnesses the power of multiple spatial datasets for identifying trajectories that have consistent patterns and are thus stable. Users can either register two (or more) tissues into a common coordinate framework and then run PSTS on the merged dataset (Fig. S13a–c). Where registration is not possible, users can run PSTS independently and identify shared driver genes that are consistently associated with common trajectories between the tissues (Fig. S13d–h); as the latter approach does not require sections to be registered, it can be broadly applied. To demonstrate the applicability of both approaches, we took advantage here of available ST data from an adjacent tissue section of the same breast cancer sample. We demonstrate that registering sections to a common coordinate framework identifies and/or confirms the common trajectories between sections (Fig. S13a–c). We further show that, even without image registration, the identification of shared driver genes and subsequent enrichment pathway analysis can support and annotate the common trajectory between sections (Fig. S13f–h). Thus, PSTS can effectively infer spatial trajectories across different sections through visualisation pattern matching, and by making use of shared transition markers. Overall, the analysis approach applied here provides the capability for predicting cancer progression, or drivers thereof, using a biopsy collected at time of diagnosis (often the only time that samples are collected). Based on the spatial changes between different cell states within a cancerous tissue, PSTS can suggest the possibility of invasion or metastasis, implying a huge translational potential that warrants further development.

Collectively, the above data validate the PSTS concept and confirm that the various constructed spatial trajectories can accurately model and/or predict biologically significant spatio-temporal changes in cell states in health and disease.

## stLearn cell−cell interaction analysis uses Spatially-Constrained and Two-level Permutation of genes and cells

Cell−cell interactions (CCI) are important in all multi-cellular processes, both for normal tissue growth or maintenance, and in disease-driven change. Current methods to find biologically significant ligand-receptor (LR) interactions in any of these contexts often suffer from a

common limitation, that is, high false discovery rates. For instance, scRNA-seq data lacks spatial context, meaning that interactions could be predicted between cell types that are spatially very distant from one another, and are thus unlikely to directly interact. stLearn's Spatially-Constrained and Two-level Permutation (SCTP) analysis solves this issue by first identifying spatial neighbourhoods of ligand-receptor co-expression, computing so-called LR scores (see "Methods" section). This is then followed by a unique constrained, two-level permutation test of both genes and spots/cells to robustly identify spatial locations where a given LR pair has significantly higher scores than random. This removes potential bias towards highly expressed genes and spatial location, thus reducing false discovery. Optionally, among the significant LRs and spatial locations, we continue to permute cell types by randomly shuffling cells/spots to different spatial locations to also test for cell type pairs that are significantly over-represented in those regions (Fig. 4a). In doing so, stLearn can make specific inferences about three important processes: cell type interactions (at the level of individual cells or spots), the LR pairs that are used for these interactions, and the spatial locations with the most active interactions in the tissue, as presented below.

To assess spatial CCI applications, we interrogated two biological systems (mouse and human), measured by three different technologies: a mouse cortex SeqFISH+ dataset (1000 genes, single-cell resolution; 4b–d), Fig. S14a–g, a mouse hippocampus dataset (Slide-seq, subcellular resolution; 4e–g), Fig. S14h–k), and a human breast cancer dataset (Visium, measuring all genes, at a resolution of 1–9 cells/spot; Figs. 4h–j and S15). stLearn SCTP identified spatially significant LR pairs in each context.

From stLearn SCTP's unsupervised analysis of spatial SeqFISH+ data of the mouse brain, we found the highest interacting significant LR pair to be Gas6-Axl in the subventricular zone (Fig. 4b–d), where it likely plays a role in regulating neurogenesis[34]. As stLearn SCTP is broadly applicable at different resolutions, it can be scaled to millions of cells using a binning strategy. The advantage of this becomes quickly apparent when stLearn SCTP was applied to a spatial Slide-seq dataset for the mouse hippocampus containing tens of thousands (47,573) of cells. Here we could reduce the run time markedly by binning cells based on their spatial location (Fig. 4e–g). The binning produced very similar results compared to stLearn SCTP analysis performed on the original single-cell resolution data (Fig. S14h–k). A similarly positive outcome was also observed when applying the same approach to the SeqFISH+ data (Fig. S14a–g).

For the Visium breast cancer dataset, stLearn identified GPC3-IGF1R as the most significant actively interacting LR pair amongst the total pool of 750 non-zero LR pairs detected within DCIS regions (Figs. 4h–j and S15). The binding of GPC3 to IGF-1R leads to downstream activation of extracellular signal-regulated kinase (ERK), which in turn induces/enhances oncogenicity[35,36]. Indeed, our enrichment

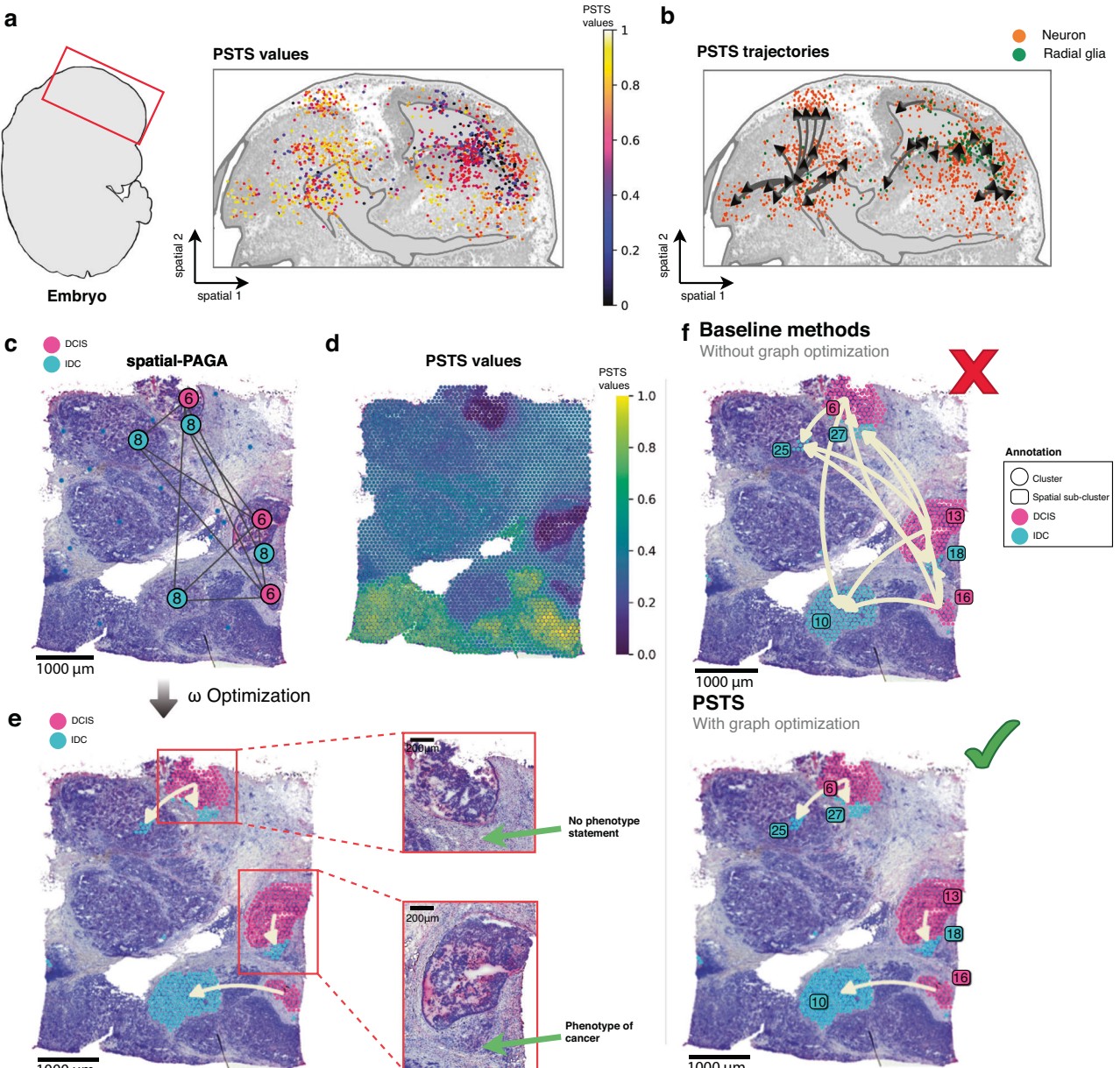

**Fig. 3 | Application of pseudo-time-space (PSTS) analysis to embryonic mouse brain development and human breast cancer metastasis. a** Mapping of PSTS values for radial glia and neurons onto the embryonic brain (sci-Space data[33], 15,466 cells). The embryonic brain region is outlined in red (left). **b** PSTS branching processes in the context of neuronal migration during brain development. Neurons and radial glia are coloured orange and green, respectively, with branching arrows indicating the developmental trajectories predicted by PSTS. **c** Spatial-PAGA graph result showing sub-cluster connectivity in a human breast cancer tissue section. **d** Visualisation of PSTS values across the breast cancer tissue array (3813 spots for one Visium breast cancer tissue section). **e** PSTS prediction of metastasis from DCIS

(ductal carcinoma in situ; pink clusters) to IDC (invasive ductal carcinoma; cyan clusters) by graph optimisation, and finding the optimal ω parameter to combine physical distance and gene expression (pseudotime; see also Fig. S10). H& E images to the right are magnifications of the two branches of the reconstructed trajectory, showing separate IDC lesion sub-clusters in different stages of invasion, with either a 'no cancer' (top) or cancer (bottom) cell appearance. **f** Non-spatial pseudotime analysis (top), suggesting non-significant and/or noisy trajectories that connect all nodes (each node is a subcluster); only PSTS can show three independent cancer progression clades (bottom).

analysis of the top LR pairs (with the highest ranks in the number of significant spots) showed a strong association with known biological processes mediated by cell–cell signalling, including a significant enrichment for the ERK1/2 cascade (Fig. S15g). When considering cell type information, we found that GPC3-IGF1R interactions were most significant between cells expressing luminal androgen receptor (Luminal-AR) within DCIS and mesenchymal breast cancer cells surrounding the DCIS regions (Fig. 4i, j). This finding hints at a potential role for this interaction in IGF1R-driven epithelial-to-mesenchymal

transition, and in itself is also in general agreement with the PSTS trajectory predictions for DCIS transitioning into IDC (Fig. 3e).

Overall, stLearn SCTP thus robustly works at different resolutions, across different scales, technologies and biological systems, to identify and rank significant LR pairs in healthy and diseased states.

Comprehensive benchmarking further showed that stLearn SCTP markedly reduces false positive predictions. Compared to existing methods, stLearn is the only method to make explicit use of the spatial location of gene expression, and also the cell types present, to predict

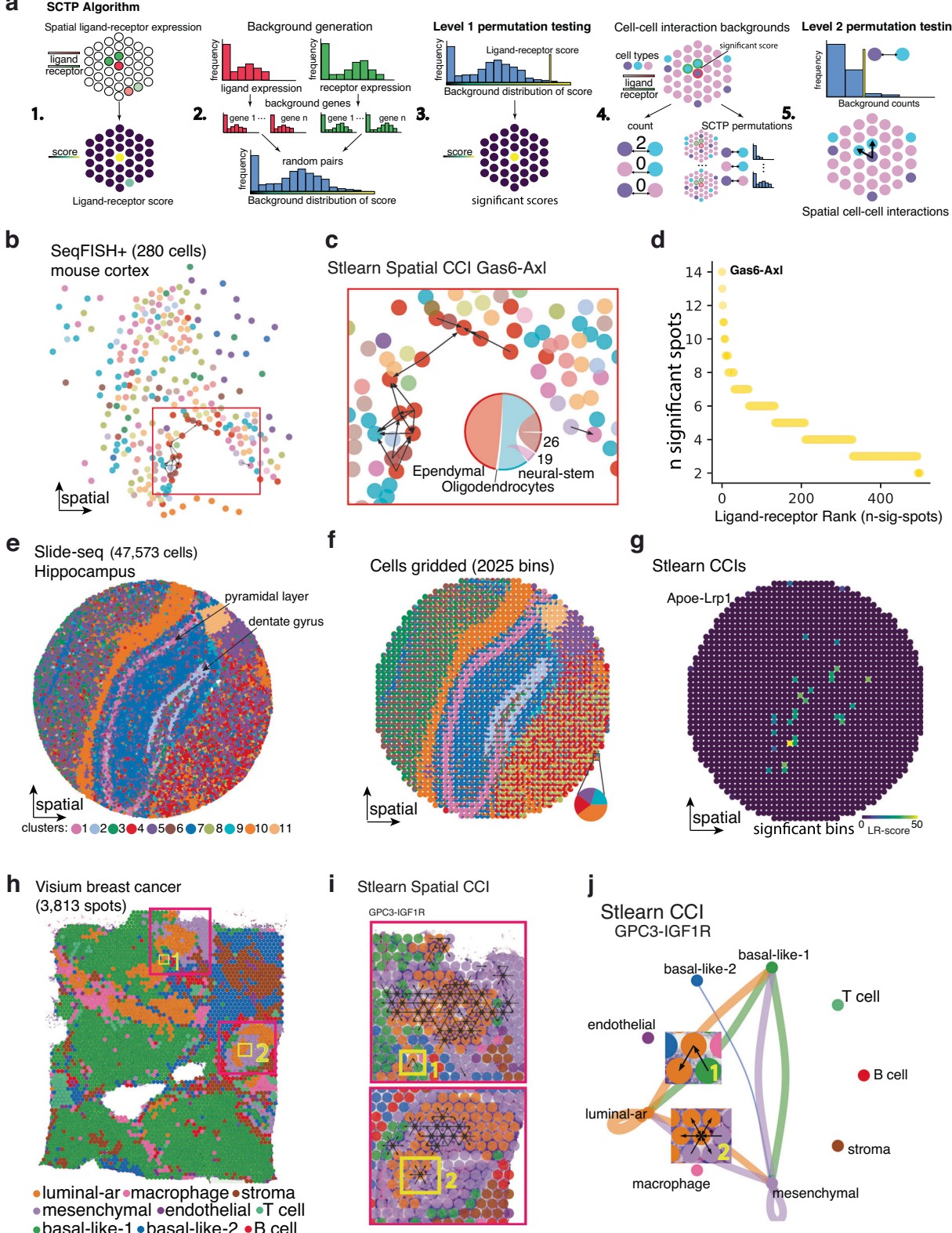

individual CCI events (Fig. 5a). Our working hypothesis was therefore that the addition of spatial information would greatly reduce the number of false positives in predicted LR interactions between cell types compared to existing methods that either do not incorporate this information, or do so less optimally, including Squidpy[37], CellPhoneDB[38], NATMI[24], SingleCellSignalR[39], CellChat[40], NCEM[41], SpaTalk[42] and spaOTsc.

To test and/or validate this premise, we first compared stLearn SCTP's performance against these eight other CCI methods using a simulated dataset; the simulation established realistic gene expression values based on either scRNA-seq or spatial data, also hypothetically arranging cellular neighbourhoods in different scenarios where individual spots represent multiple cell types (see "Methods" section and Fig. 5b). Compared with the ground-truth for cell–cell interactions in

**Fig. 4 | A Spatially-Constrained Two-level Permutation (SCTP) test for cell–cell interaction (CCI) analysis. a** Overview of the stLearn SCTP algorithm, which uses spatial location and ligand-receptor (LR) co-expression to predict interactions in multiple spatial technologies: (1) spatial neighbourhoods are scored for LR co-expression, (2) background spatial co-expression is determined by randomly pairing genes (default 1000 pairs) with equivalent expression levels to LR pair, (3) significant spots of spatial LR co-expression are determined by comparison to the random background, (4) counting of cell type co-occurrence in neighbourhoods of significant LR co-expression, with and without permutation of cell type information, and (5) cell types with significant co-localisation in regions of LR co-expression are predicted as interacting. **b** stLearn SCTP results for the top-ranked LR pair *Gas6-Axl* in seqFISH+ data from mouse cortex. **c** Enlarged panel of the boxed area in **b**, showing the subventricular zone; black arrows connect interacting cells, and chord plot summarises predicted CCIs facilitated by *Gas6-Axl*. **d** Scatter plot highlighting the top predicted LR pair by stLearn SCTP (*Gas6-Axl*), with the number

of significant cells on the *y*-axis and LR pairs on the *x*-axis. **e** Mouse hippocampus Slide-seq data annotated by cluster. **f** Cells binned by spatial location, with bins representing mixtures of cells similar to Visium data. Bins are represented as pie charts showing the breakdown of cell types. **g** Significant co-expressing spots for the top-ranked ligand-receptor pair *Apoe-Lrp1*, illustrating that SCTP can scale to a large number of cells by binning. **h** Visium ST data from human breast cancer, with each spot coloured by the dominant cell type, as predicted by deconvolution. Red boxes correspond to Ductal Carcinoma In Situ (DCIS), and yellow boxes show regions highlighted in **i** and **j**. **i**, DCIS regions showing significant SCTP predictions for a highly-ranked LR pair (*GPC3-IGF1R*), overlayed as arrows, where the receiving spot expresses the receptor and the output spot expresses the ligand. **j** Network diagram of SCTP-predicted CCI results for *GPC3-IGF1R*. Zoomed-in images of interacting spots (from yellow boxes 1 and 2 in **h** and **i**) are shown on the edges, connecting relevant cell types in the graph.

the simulated dataset (Fig. 5c), stLearn was the only method able to reconstruct these interactions without any additional false positive interactions (Fig. 5d). This example highlights that for ST data not at single-cell resolution (e.g. Visium), methods that do not take into consideration that each spot may be a mixture of cell types can notably misrepresent the association of gene expression with the cell type information, consequently predicting many false positive interactions between cells and/or cell types.

Methods that do not consider spatial information are otherwise also likely to predict interactions between distal cell types. Any such predictions are unlikely to represent true interactions, as these generally occur within a range of 200 μm[43,44]. We therefore benchmarked stLearn's performance against other CCI methods using the experimentally generated Visium breast cancer dataset (Fig. 5e). When using cluster information as the input into either stLearn or the other CCI methods, only stLearn correctly predicted no interactions between distal clusters (Fig. 5e-f). When using cell type information for each breast cancer spot, stLearn SCTP again performed significantly better than existing methods (Fig. 5g–l). The stLearn SCTP pipeline otherwise also provides a means to extract significantly interacting spots/cells, which allows for downstream analysis of LR pathways to further validate the predicted interaction (Supplementary Note 2 and Fig. S16). Overall, we find that stLearn's SCTP algorithms significantly outperform all existing CCI methods in terms of resolution and biological plausibility.

RNAscope imaging produced evidence supporting the predicted CCI events. Specifically, to assess stLearn's ability to detect individual CCI events, we first tested an interacting LR pair, *IL34* and *CSF1R*, which is known to be active in multiple cancer types[45] and was previously shown by us to have potential immunoregulatory roles in skin cancer[46]. Here, we therefore generated ST data (using the 10x Visium protocol) for human basal cell carcinoma (BCC) skin cancer samples and then applied stLearn SCTP to detect IL34-CSF1R interaction events across the tissue (Figs. S17 and S18). For experimental validation, we then used an adjacent tissue section from the same BCC tissue block for RNAscope analysis[47]. We selected RNAscope for independent validation because it is a single-cell resolution imaging technology, capable of detecting messenger RNA molecules at single-molecule sensitivity (Fig. S18). We performed image registration here to align RNAscope and Visium data, allowing us to compare the results between the two orthogonal technologies. If interactions predicted by stLearn's SCTP analysis of Visium data were correct, then we would expect to see co-localisation of these LR genes with RNAscope in the same regions of the adjacent tissue section. Image registration matching between RNAscope spots and stLearn's predicted CCI events (*IL34* and *CSF1R*) indeed showed consistent correspondence between the predicted and observed interaction events, mostly at the border between cancer nests and normal tissue areas (Fig. S18c).

## stLearn imputes missing data and corrects for technical variation across the tissue

In general, single-cell and spatial PCR-based sequencing technologies suffer from dropouts, that is, the misdetection of lowly expressed genes due to suboptimal capturing efficiency (or lack thereof) with a small amount of starting material. Further, spatial sequencing has tissue regions that are not measured. For example, Visium sequencing data has a space between two spots, and most other spatial technologies measure only selected regions of interest, leaving others uncaptured. stLearn introduces an imputation method that can address both of these limitations. The assumption for this imputation method is that missing information in one spot can be rescued and/or corrected based on reference spots that are highly similar based on tissue imaging data (i.e., tissue morphology; matrix I), are spatially close (similar X, Y coordinates in spatial data; matrix D), and/or have similar expression profiles across all genes (e.g. Pearson correlation; matrix G) (Fig. 6a). Indeed, imaging data alone carries information of functional significance across the tissue, as exemplified for both the breast cancer (Figs. S19 and S20e) and mouse brain (Fig. S20a–d) ST datasets. We provide evidence here that there is a clear added benefit for stratifying tissue regions when morphological image features are included and integrated with both gene expression and spatial distance information (Fig. S19). In addition, we also demonstrate the 'value add' of the ResNet50 neural network model over simpler forms of matrix I that use handcraft rather than ResNet50 image features as the morphological data input (Fig. S20a, b). Here, handcraft image features were not able to define specific regions as specific and/or accurate as the ResNet50 model (Fig. S20c–e). Guided by these findings, we applied a "pooled spot reference" approach to develop an imputation method (hereafter referred to as stSME, see below), where ResNet50 features for cell/tissue morphology, spatial distance and transcriptional data are all taken advantage of to correct for possible 'dropout' (0 values), and to predict gene expression in intermediate tissue regions (pseudo-spots) not covered by a spatial spot and that are thus not measured (Fig. 6a).

We validated this method by simulation, with a 'leave-out' validation strategy where gene expression from the original data was randomly set to zero and then corrected by *S*patial *M*orphological gene *E*xpression (stSME) adjustment. We show that the stSME approach was able to correctly recover the 'leave-out' (and 'dropout') data, and also that the imputed data significantly improved overall clustering accuracy compared to the dataset without imputation (Fig. 6b, d), or when handcraft image features were used instead of those extracted by ResNet50 (Fig. S21a, c). Our imputation approach allowed for specific sub-regions of the hippocampus to be resolved and separated here, that is, the Cornu Ammonis 1 (CA1) region (cluster 6, Fig. 6d) from the CA3 region (cluster 17); non-imputed data failed to detect this region (Fig. 6d), as did handcraft image features (Fig. S21a) and many of the other methods that we benchmarked against using

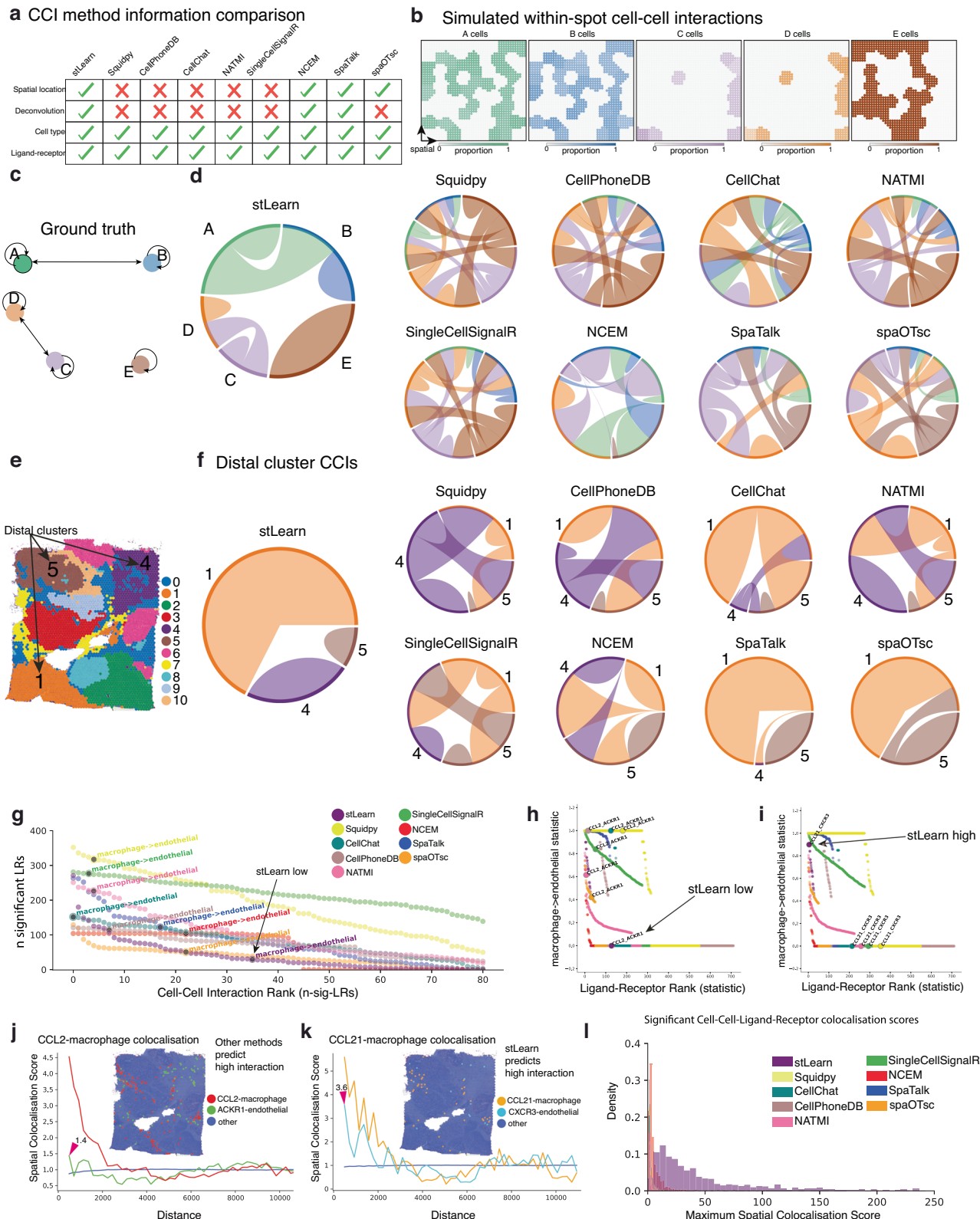

the same cluster resolution (Fig. S22a–d). We further show that, when applied to Visium data, stSME did not skew the distribution of the total counts per spot but benefited analysis outcomes by increasing the number of non-zero values (Fig. 6c). The increased performance after stSME imputation was directly evident from higher adjusted Rand index values (Fig. 6e; 10 bootstrap analyses on simulation data in d), and it was also shown to be robust across biological replicates (Fig. 6f). Because stLearn's stSME strategy is based on carefully (but

automatically) selected reference spots, it additionally circumvents the over-smoothing and/or over-correction issues that are often seen in spatial smoothing methods. This is exemplified by the enhanced detection of marker genes that are specific for the CA3 (*Lhfpl1* gene) and dentate gyrus regions (*Pla2g2f* gene), respectively (Fig. 6b); this feat was made possible by ResNet50 but not handcraft image features (Fig. S21b). We corroborated this by examining a ST dataset from human cortex, where stSME once again showed a high overall level of

**Fig. 5 | stLearn's Spatially-Constrained Two-level Permutation (SCTP) analysis reduces false positive predictions and enriches for co-localised cells expressing LR pairs. a** Summary of information utilised by stLearn SCTP and eight other methods (used for benchmarking) to predict cell–cell interaction (CCI) events. **b** ST data simulation with multiple cell types per spot. Five cell types, named A to E, are shown with pair-wise co-localisation of A and B, C and D, contrasted by the exclusion of E. **c** Ground truth of CCIs for simulation shown in **b. d** Chord plots representing predicted CCIs by stLearn, Squidpy, CellPhoneDB, CellChat, NATMI, SingleCellSignalR, NCEM, SpaTalk and spaOTsc. Only stLearn predicts the ground-truth without false positive interactions. **e** Visium ST data for human breast cancer with spots coloured by cluster IDs. Spatially distant clusters 1, 4 and 5 are high-lighted. **f** Chord plots showing predicted CCIs by stLearn SCTP and benchmarking methods. **g** Scatter plot showing the number of significant LRs for each cell type combination (81 from 9 cell types) on the *y*-axis and all pairwise cell–cell combi-nations on the *x*-axis, ranked by the number of CCI interactions per pair. The

'macrophage to endothelial cell' interaction is highlighted as an example where stLearn correctly ranked it low. **h** Scatter plot showing the statistic for 'macrophage to endothelial cell' interactions (scaled between 0 and 1 for comparison) on the *y*-axis, and the ranking of LR pairs on the *x*-axis. *Ccl2-Ackr1* is highlighted as an example where only stLearn correctly predicted no interactions. **i** Same as **h**, but highlighting a different LR pair (*Cxcl21-Cxcr3*), predicted by stLearn SCTP (but not other methods) to be involved in macrophage and endothelial cell interactions. **j** Co-localisation results (spatial distance) for *Ccl2*-expressing macrophages and *Ackr1*-expressing endothelial cells (refer to **h**). Co-localisation scores are on the *y*-axis and neighbourhood distance from the *Ccl2*-expressing macrophage on the *x*-axis. **k** Equivalent to **j**, except that the LR pair *Cxcl21-Cxcr3* from **i** is shown. **l** Histogram of maximum co-localisation scores across all cell types and the top-50 LR pairs facilitating interactions between these cell types; stLearn exhibits an overall increase in spatial enrichment for predicted CCIs.

performance (Fig. S22e), with stSME-based clusters also not showing the over-smoothing artefacts commonly seen with other methods at the same clustering resolution (Figs. S22f and S23). Importantly, as alluded to earlier, stLearn's stSME algorithm can also be successfully applied to predict gene expression for tissue regions with no experi-mental information (such as gaps between Visium capture spots) using both morphological similarity and physical distance (Fig. 6g). This way, stSME computationally increases tissue coverage of ST data, which otherwise leaves unmeasured gaps between spatial spots and/or regions of interest (Fig. 6g). Taking all this together, we posit that our stSME approach will remain important, even as spatial transcriptomics technology continues to advance towards higher-resolution mea-surements, as technical dropout issues are likely to be more proble-matic here (i.e., increased resolution at the expense of reduced sensitivity).

Overall, stLearn's integrative stSME analysis approach results in enhanced data quality and significant improvements in clustering, as indicated by the accurate segregation of anatomical sub-structures and/or cell types in the tested brain and breast cancer datasets (Figs. 6d, g; S22; and S24). As shown, the stSME method corrects for dropouts and technical variation in spatial sequencing data, using both imaging and spatial information to predict gene expression. Notably, even for marker genes that are known to be highly expressed in spe-cific regions (e.g. *Pla2g2f* and *Lhfpl1* for the dentate gyrus and CA3 regions of the hippocampus, respectively), there were spots with a '0' expression value (dropout - Fig. 6b, left), and these values could be rescued by stSME imputation (Fig. 6b, right). This proves imputation useful, not just for low- but highly expressed genes also.

## Discussion

We have developed and validated three spatially-guided algorithms and analysis tools to address unmet needs around the processing of ST data, namely finding dynamic trajectories of biological processes within a tissue section, a means to robustly detect cell–cell commu-nication in situ and, lastly, for dealing with dropout issues and/or data sparsity. stLearn's comprehensive analysis toolkit was purposefully developed in such a way that it can be applied to a wide range of biological settings, and also for applications where only spatial loca-tion and gene expression information is available. Optimal stLearn performance is achieved, however, when tissue morphology informa-tion from an H&E image (or other stain) is also included. That said, even without such imaging data, e.g. as with Slide-seq[15], MERFISH[14], seqFISH[13] and sci-Space data[33], stLearn is still able to accurately infer spatial trajectories and cell–cell interactions based on spatial data and gene expression information alone.

Our PSTS algorithm for inferring trajectories in ST datasets advances from existing methods (which were mostly developed for scRNA-seq)[29,48,49] by adding the spatial dimensionality that accounts for the similarity of neighbouring cells. PSTS also uniquely adds the

ability to trace spatial branching processes[50] via trajectory inference. In doing so, PSTS outperforms other pseudotime methods[29,51] that were not designed for spatio-temporal modelling tasks. Overall, PSTS pro-vides users with an overview of dynamic processes occurring across two or more anatomically and/or morphologically defined regions, and we showed its broad applicability in various biological or medically relevant processes and conditions that are known to evolve across both time and space. We also tested that our PSTS analysis works for technologies at both single-cell (sci-Space dataset) and multi-cell (e.g. Visium) resolution. By balancing spatial with gene expression infor-mation, our PSTS method can also help solve trajectories when one of the intermediate states may be missing. Specifically, where an inter-mediate node or state is missing from the section (i.e., when there is no direct and/or obvious spatial connection), the relatedness of other spots / nodes in the trajectory can still be inferred from the global transitional pattern within the gene expression information (as observed in PCA/UMAP/Speudotime latent space). Lastly, for instances where integration of data from multiple sections and/or conditions is required (a major bottleneck in the field that is beginning to be addressed[52]), we offer the following recommendations and/or work-arounds for spatial trajectory analysis. For adjacent sections (replicates from the same tissue block), these can be transformed into a common coordinate framework to form either a large 2D spatial array by expanding the original matrix, or a 3D spatial matrix by adding layers into the original data. This merged dataset can then be used directly for stLearn's PSTS trajectory analysis (Fig. S13c). For sections that cannot be transformed into a common coordinate framework, such as those representing samples from different biological and/or treatment conditions, we suggest applying the spatial trajectory method to each section first and then finding the shared transition markers between trajectories (Fig. S13f). By calculating the intersection proportion between two (or more) sections based on these markers, it is then possible again to identify and/or annotate the common trajectories as well as to independently confirm their existence.

The ability to study cell–cell interactions is key to understanding complex tissue ecosystem dynamics. stLearn's spatial statistical test was specifically designed to reduce the problem of high false-positive detection rates that CCI analysis typically suffers from. We achieved this by using spatial constraints, and also by removing potential bias towards abundant LR pairs and/or spots/cells with overall high expression of most genes (Fig. 4a and S15). Existing CCI methods such as CellPhoneDB[38], CellChat[40], NATMI[24] and SingleCellSignalR[39] were all originally developed for scRNA-seq data and do not use spatial information[44,53]. More recent spatial-based methods like NCEM[41], SpaTalk[42] and spaOTsc also appear to not fully exploit and/or use the spatial factor in CCI analysis, as they all still produced false positive interactions between distal cell types that are unlikely to occur in both simulated and experimental datasets (Fig. 5). stLearn's SCTP analysis stands out in that it simultaneously uses the spatial distribution of

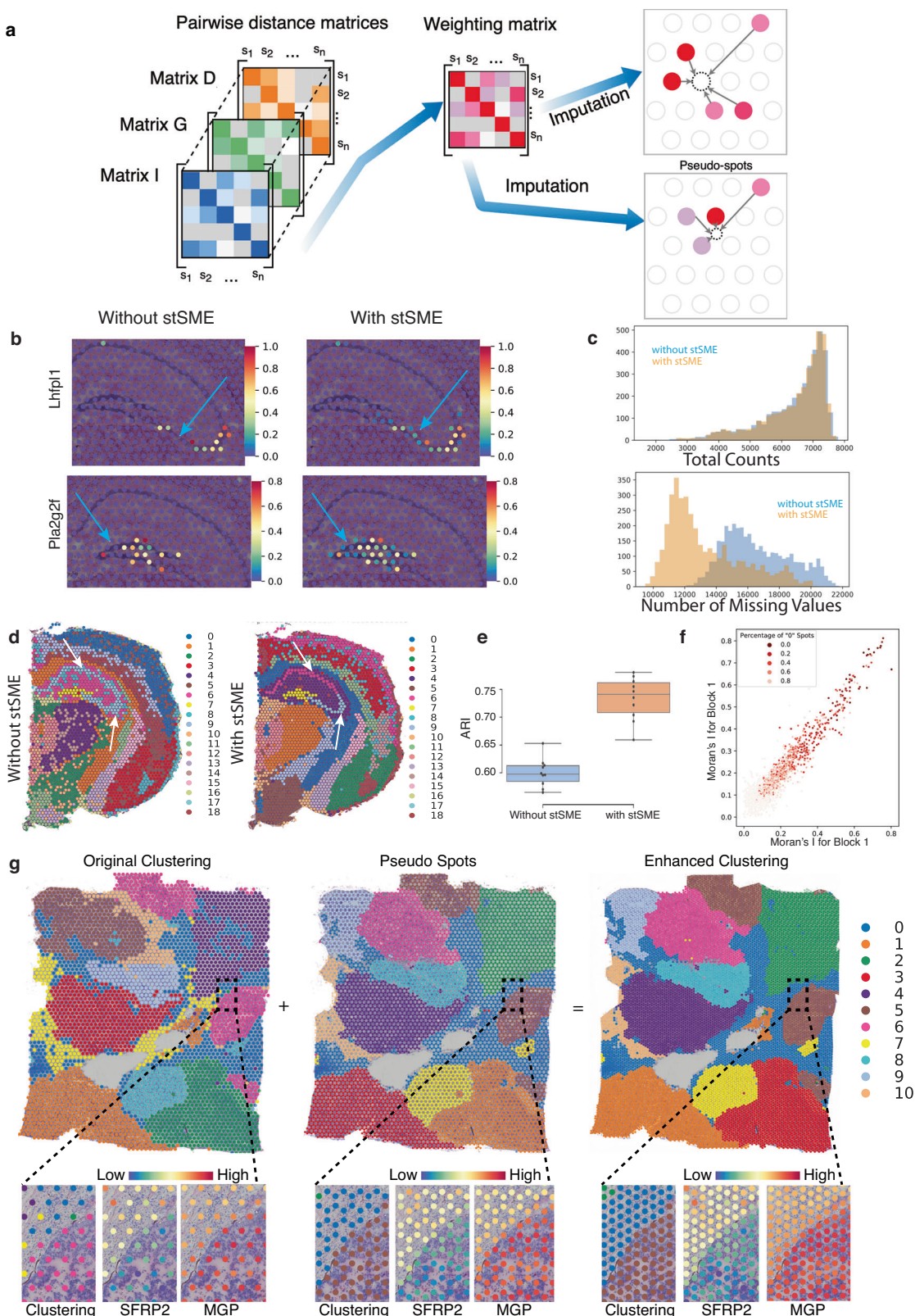

neighbour cells, gene expression, and prior knowledge of LR interactions. It assumes significant interactions to be dependent on the cell type[13,23] and non-random co-expression of the LR pair among neighbouring cells within a range of 200 μm[43,44]. stLearn also differs from other methods (e.g. DIALOGUE[54]) that focus on identifying coordinated gene expression programs in different cells based on regulated genes, but that do not directly test for significant interacting LR pairs; the multicellular programs used by DIALOGUE only include genes that are up- or downregulated in different cell types, and these may or may not be LR pairs. Overall, comprehensive benchmarking and independent validations demonstrated the ability of stLearn to (1) detect individual CCI events in ST data, (2) reduce false positives for predicted CCI events, and (3) identify spatial regions with biologically important interactions, e.g., cancer-immune cell crosstalk (Fig. 5).

**Fig. 6 | Application of stLearn stSME imputation to spatial datasets with morphological information. a** Schematic showing stSME integration of three data types (imaging morphology (I), gene expression (G) and spatial location/distance (D). stSME finds biologically relevant reference spots, to then adjust existing spots, or predict gene expression for new spots (pseudo-spots) by imputation. **b** Rescue of dropout (zero values; blue arrows) by stSME for gene markers of the Cornu Ammonis (CA) 3 (*Lhfpl1*) and dentate gyrus (DG; *Pla2g2f*) regions of the mouse hippocampus. Note that the imputation is specific to biologically relevant spots. **c** Effects of imputation on library size (total gene counts per spot; top), and the number of spots with missing values (bottom). **d** Simulation approach assessing stSME imputation performance using mouse brain Visium ST data. Louvain clustering was performed with imputed values after randomly removing 20% of values from the original (log transformed UMI counts) data as a 'leave-out' validation strategy. Note that clusters without stSME imputation are much noisier, and also that the hippocampal CA1 (cluster 6) and CA3 (cluster 17) sub-regions could not be separated (white arrows). **e** Box plot showing poorer clustering results when stSME is not used, as assessed by adjusted Rand index (ARI; data was randomly sub-sampled 80% from 2702 spots of a brain section, with a total of $n = 10$ simulations). ARI was calculated using the full data clustering results as the reference. **f** Robustness and performance of stSME imputation method for the top-2000 highly variable genes (HVGs) across two replicate sections of the Visium human breast cancer ST dataset (10x Genomics; Block A, sections 1 and 2; see "Methods" section for details). Data points are the spatial autocorrelation (Moran's I index) for the same set of imputed HVGs in section 1 (*x*-axis) and section 2 (*y*-axis); colour coding reflects sparsity of the gene in the original UMI count matrix. **g** Imputation of gene expression in regions without data (i.e. array gaps) improves tissue coverage and clustering in human breast cancer samples. Bottom images show zoomed-in displays of boxed DCIS boundary region, showing cluster location and expression of breast cancer markers *SFRP2* and *MGP* (abundant in DCIS).

stLearn also implements an imputation method (stSME) to improve the quality of noisy and/or incomplete spatial sequencing data (Fig. S21). Specifically, stLearn uses an interpretable model to integrate morphological similarities with physical distance and gene expression similarities (Figs. 1e, S19 and S20). This integration effectively deals with the issue that lowly expressed genes are often either not detected or have high technical variation (Fig. S21). Our stSME method is based on the quantitative link between cellular morphology (e.g. cell size, nuclei size, granularity, density, or distribution) and molecular gene expression profiles[16,19,55–57], with the innovative aspect being that imaging data is used to correct and/or impute the sequencing data. Models like SpaCell[17] have already shown that combining imaging pixel information and gene expression more accurately classifies cell types than other models using either gene expression or tissue image data alone. stSME advances by also utilising spatial distance, based on the known positive correlation between gene expression and proximal physical distance[21,22,58]. By selecting reference spots based on imaging, gene expression and spatial locations, we showed stSME to be capable of reducing missing values and technical variation, but without undesirable 'smoothing' effects. Indeed, after stSME imputation, the number of non-zero spots is higher but not all spots become non-zero, indicating that the approach is not over-correcting. Overall, our stSME procedure leads to highly desirable outcomes and/or effects, including: (1) the recovery of 0 values, where the distribution of total read counts per spot is preserved but in which the number of spots with 0 values for each gene is reduced (Fig. 6c); (2) increased sensitivity for differences between spots that belong to different cell types and/or regions (Figs. 6d and S22); (3) improved downstream clustering performance after stSME imputation (Figs. 6g and S21); and (4) no over-smoothing or global effects due to local outlier spots (Figs. S22d, f and S23). The integrated stSME approach also enables the inference of gene expression where there are no spatial spots and/or cells being measured, a beneficial application when tissue regions need to be considered more comprehensively (Fig. 6g); as shown, these predictions are based on and/or derived from the H&E image and neighbouring spot(s) with high morphological similarity to the unmeasured region of interest.

Lastly, while our advancements cater to various aspects of ST data analysis, we acknowledge that some (inherent) limitations remain. Specifically, the stLearn toolkit offers substantial utility for detecting dynamic trajectories of biological processes within tissue sections, and for identifying cell–cell communication. However, it presently remains constrained by the fact that ST data stems from thin tissue sections of 5-10 micrometre thickness (2D space). The absence of a true third dimension currently restricts the ability to capture the full complexity of cellular arrangements within tissues, particularly in cases where interactions may occur across multiple tissue planes. In addition, longitudinal datasets with a temporal dimension are limited. Our computational modelling utilises the spatial variation in transcription and tissue morphology across the tissue section, which partially represents the trends in gene expression in both spatial and temporal axes. Going forward, however, new experimental methods that better address these limitations will be imperative to harness the complete potential of spatial transcriptomics, and to advance our understanding of the intricate dynamics of cells within their native tissue contexts.

Taken together, the three methods presented here have all been tested across a wide range of biological systems and applications, were validated both computationally and experimentally, and also benchmarked against existing methods. We built stLearn as one of the very few Python-based platforms for spatial and imaging data analysis, implementing algorithms like PSTS, SCTP and stSME, and offering fast computation. Detailed tutorials and documentations are available to ensure reproducibility and ease of use. Importantly, we also produced an interactive version of stLearn (i-stLearn; see Software Implementation section), allowing experimentalists to make use of this powerful analysis platform without the need for coding. Because of this, we expect stLearn to become a valuable analysis suite for utilising the exponentially growing amount of ST datasets, and we will be actively maintaining and developing the software, including the interactive web-based application, to ensure its currency and continued usage by the broader community.

## Methods

All animal experiments were conducted in accordance with the Australian Code for the Care and Use of Animals for Scientific Purposes, and with approval from The University of Queensland Animal Ethics Committee. The work that involved the patient sample reported here (i.e., skin biopsies from the patient with basal cell carcinoma) was reviewed and approved by Metro South Human Research Ethics Committee and by The University of Queensland Human Research Ethics Committee (HREC-11-QPAH-477, The University of Queensland, Clearance No. 2012000052). Informed consent was obtained from the patient participant.

### Mouse TBI datasets
**Experimental model details.** Three-month-old female C57BL/6J (Animal Resources Centre, Canning Vale, WA, Australia) and $CX_3CR1^{creERT2}$ x iDTR x tdTomato mice (bred and maintained in a University of Queensland Biological Resources specific pathogen-free 'behind barrier' facility) were used in this study. Experimental mice were housed socially (3−5 mice per cage) on a 12-h light-dark cycle in individually ventilated cages, with adlibitum access to food and water.

**Visium ST library preparation.** Mice were subjected to controlled cortical impact injury or sham surgery (i.e. craniotomy only) as described previously[25]. Injury parameters were: impact speed, 3.5 m/s; deformation depth, 1.0 mm; duration, 400 ms. Mice were sacrificed three days after TBI or sham surgery, their brains dissected in an RNase-free environment and immediately transferred into refrigerated O.C.T. compound (Sakura Tissue-Tek O.C.T. compound, Nagano, Japan) for

flash-freezing in chilled isopentane. Samples were stored at −80 °C until further processing. Library preparation of mouse brain samples was performed according to the Visium Spatial Gene Expression Reagent Kits User Guide (CG000239 Rev C, 10x Genomics, USA). Briefly, brain samples were cryosectioned until the dorsal hippocampus and dentate gyrus region were visible. Next, 10 μm sections were collected onto a pre-chilled Visium slide. All sections were dried onto the slides at 37 °C for 1 min, fixed in pre-chilled 100% methanol at −20 °C for 30 min, and stained in Mayer's Haematoxylin for 5 minutes and Eosin for 2 minutes. Slides were then mounted in 85% glycerol for coverslipping and brightfield imaging (Axio Z1 slide scanner, Zeiss). Permeabilisation of mouse brain samples was carried out for 18 minutes, resulting in a cDNA library size of 470 bp post-fragmentation. Library quantification was carried out using the KAPA Library Quantification kit (Roche), followed by in-house sequencing using a high output reagent kit and NextSeq500 instrument (Illumina) at the Institute for Molecular Bioscience Sequencing Facility. Sequencing was performed using the following protocol: Read1 - 28bp, Index1 - 10bp, Index2 - 10bp, Read2 - 120bp. Raw BCL files were processed by bcl2fastq V2.7.0, and the fastq reads mapped to the mouse reference genome GRCm38 by SpaceRanger V1.0.0.

**Legacy ST library preparation.** Additional TBI samples (6 h and 3 days post-injury) were independently prepared with the Legacy ST kit, an earlier version of 10x Genomics' ST platform. Mouse TBI samples were obtained as described above and prepared for ST analysis following the Library Preparation Manual Version 190219 (10x Genomics, USA). Briefly, ipsilateral brain hemisphere samples were collected at 6 hours and 3 days post-TBI and embedded in O.C.T. Next, 10 μm cryosections were collected onto the Legacy ST library preparation slide, fixed with 4% paraformaldehyde at room temperature and then stained with haematoxylin for 10 min, blueing buffer for 1 minute, and eosin Y for 3 min in an RNase-Free environment. High-resolution H&E images were again captured using the Zeiss Axio Imager. ST sequencing libraries were prepared as per the manufacturer's instructions, with pre-permeabilisation and permeabilisation performed at 20 minutes and 7 minutes, respectively. On-slide cDNA synthesis, tissue removal, probe cleavage and final library preparation were all performed as per the manual. High-quality cDNA libraries with sizes ranging between 660-780bp were obtained and sequenced on the Illumina Nextseq500, using a 150 cycles kit with read configuration as read 1 (26 bp) and read 2 (124 bp).

**Sequencing data pre-processing.** The 3 days post-TBI Visium sample was used for spatial trajectory inference, while the three other mouse brain ST samples (i.e. Visium sham control, Legacy 6 hours post-TBI, and Legacy 3 days post-TBI) were used to validate PSTS results at different time points, conditions and/or technological characteristics. All subsequent references to the TBI sample refer to the 3 days post-TBI Visium sample used for PSTS analysis unless otherwise specified. The raw data consisted of 2442 spots within the tissue area and 20,787 genes with a median of 4264 genes per spot. We filtered low-quality data by removing spots expressing fewer than 200 genes (i.e. spots with low transcriptome diversity) and genes expressed in fewer than three spots (i.e. genes that were too lowly expressed to reliably detected with sufficient statistical power). We then followed the standard stLearn pre-processing workflow to detect 3884 highly variable genes, normalise the counts per spot, perform log count transformation and scale gene counts to unit variance. Across all mice used, we obtained a total of 6337 spots (5410 spots from Visium and 927 spots from Legacy protocols).

**Clustering of TBI dataset.** PCA and standard Louvain clustering using the top 50 PCs were used to detect 15 broad clusters across the TBI sample (Figs. S2 and 2c). Clusters were split further if they were spatially separated within the tissue using the stLearn sub-clustering option. Each cluster was annotated using the well-defined anatomical regions given by the Allen Mouse Brain Atlas[28]. Data were visualised in both UMAP and ForceAtlas2 space[59] (Fig. S2).

**Pre-processing: filtering of microglia-related spots and genes for PSTS analysis.** Prior to PSTS analysis (described below), we performed two data filtering steps in order to focus our analysis on microglia-specific changes. First, we filtered the dataset to include only those spots that contained microglia, using the key markers *Fcrls* and *Tmem*119. Although it is possible that the selected spots also covered additional non-microglial cells, this filtering step removed spots without any microglial gene signature. Importantly, we also limited the genes used for PSTS analysis to 1998 microglia-specific genes that we previously identified in a publicly available RNA-seq dataset[25], thus minimising and/or removing any potential confounding contributions of non-microglial cells to the transcriptional signature of selected spots.

### Human skin cancer dataset

**Collection and preparation of tissue sample.** Skin biopsy samples from patients diagnosed with basal cell carcinoma (BCC) were collected at the Dermatology Department of the Princess Alexandra Hospital in 2019. Samples smaller than 1 cm x 1 cm in size were snap-frozen prior to embedding in Optimal Cutting Temperature (O.C.T.) compound for solidification. The embedded tissues were cryosectioned at 10 μm thickness, processed for ST-seq as per the TBI datasets described above, and/or transferred to a −80 °C freezer for future RNAscope Hiplex assay analysis.

**ST sequencing and data pre-processing.** These steps were performed as per the TBI datasets described above. We obtained a total of 1179 spots with a median of 1205 genes per spot.

### sci-Space mouse embryonic brain dataset

We downloaded the count matrix, spatial data and metadata of the sci-Space mouse embryo brain dataset[33] from the National Center for Biotechnology Information (NCBI) under the accession number GSE166692. From the raw data of 121,365 cells (average of 2514 UMIs and 1231 genes per cell), we subsetted to 15,466 cells in the categories "Neuron", "Glial Cells" and "Radial glia" (keeping cells with at least 200 genes but fewer than 7000 genes).

### Visium human breast cancer dataset

We obtained the Human Breast Cancer Visium dataset from the 10X Genomics website (https://support.10xgenomics.com/spatial-gene-expression/datasets/1.0.0/V1_Breast_Cancer_Block_A_Section_1. It contains 3813 spots under tissue with a median count of 17,531 UMI per spot, which equated to a median gene count of 5394 per spot.

### seqFISH+ mouse brain sub-ventricular zone dataset

seqFISH+ data[13] of the mouse brain (2963 cells) with an average of 3338 genes per cell were downloaded (https://github.com/CaiGroup/seqFISH-PLUS/blob/master/sourcedata.zip) (accessed February 2022).

### Slide-seq mouse hippocampus

Slide-seq data[15] for the mouse hippocampus containing 47,573 cells and 20,572 genes were downloaded from website (https://www.dropbox.com/s/cs6pii5my4p3ke3/mouse_hippocampus_reference.rds?dl=0), (accessed February 2022).

### Visium coronal mouse brain dataset

We downloaded the count matrix, annotation and spatial data from 10x Genomics' public Mouse Brain Visium dataset [https://support.10xgenomics.com/spatial-gene-expression/datasets/1.1.0/]. In total, there are 2702 spots, with a median of 28,944 UMIs and 6018 genes detected per spot.

### Visium human brain dataset

We downloaded the count matrix, annotation and spatial data from Maynard et al.[11] [https://github.com/LieberInstitute/HumanPilot]. There were 47,681 spots (12 samples), with an average of 3462 UMIs detected per spot, and an equivalent of 1734 genes per spot.

### Simulated datasets

In addition to the various in-house and public datasets described above, we also developed a generative approach to simulate ST data in an in silico tissue to assess cell–cell interaction methods. The simulation takes into account cell type-specific gene expression distribution, zero proportions, cell type proportions and spatial cell communities with differential co-localisation, or with specific exclusion of cell types across the tissue. This generative in silico tissue allowed us to test assumptions about contributions of spatial distance, cell type heterogeneity and false discovery in cell–cell interaction analysis results. Briefly, we initiated the simulation process by estimating the gene expression distribution for each gene in each of 11 cell types in a reference scRNA-seq dataset by fitting a negative binomial distribution[60]. Using scRNA-seq rather than spot-level data (as in Visium ST) allowed us to simulate gene expression at single-cell resolution, which can then be grouped into spot level at different mixing proportions. For each cell type, we estimated gene expression for 10,000 cells, with 10,335 genes for each cell, including top variable genes in scRNA-seq data and all known ligand/receptor genes. For each gene, a proportion of zero counts was calculated using scRNA-seq data and this was used for sampling zero values in the simulated cells to maintain sparsity. We then initialised an empty tissue with $(x, y)$ coordinates of $n$ spots evenly placed on the in silico tissue. Based on the coordinates, the spots were clustered into $\sqrt{n}$ pools. Pools that are within a distance were grouped into neighbourhoods (communities). Allocations of communities with differential combinations of cell types per pool were arranged either randomly (for null distribution), or with co-localisation and exclusion priors, such that dominant cell types for each neighbourhood were either adjacent (co-localised) or distant (exclusion) from each others. The cells simulated as above were then randomly assigned into spatial spots in each neighbourhood by sampling from a Gaussian distribution, with the mean as the proportion of that cell type in scRNA-seq data and with a user-selected variance that reflects cell type variation across spots. These generative tissues were then used to evaluate cell–cell interaction results as described below (see "stLearn cell–cell interaction analysis").

### Pseudo-time-space algorithm - spatial trajectory inference

**Spatial trajectory concept.** Spatial trajectory inference incorporates both gene expression and spatial information to infer the order of transcriptomes (i.e. spots or cells) along a trajectory, thereby allowing the spatio-temporal pattern of a given dynamic process to be revealed. The addition of spatial dimensionality information distinguishes this method from traditional pseudotime approaches used with scRNA-seq data analysis[29,51]. Specifically, spatial trajectory inference allows for detailed mapping of branching phenomena representing spatial motions within a tissue across time, such as cell activation, differentiation or cancer evolution (see Supplementary Note 1). In this concept, we used the node to represent the cluster/sub-cluster of cell types, with the branch not only giving the spatial direction but also a summary of the change of gene expression along it.

**Selection of the root.** Unlike trajectory analysis in scRNA-seq data, the selection of the root spot in a spatial trajectory for ST-seq data is more dependent on its physical location. We provide an option to semi-automatically select the root spot. First, users need to define the cluster that could be the root cluster (initial state). Next, the CytoTRACE[61] scoring system is applied to calculate the number of genes expressed per spot (*num_exp_genes*). After that, we rank genes based on the correlation between their gene expression and the

*num_exp_genes*. Then, top-correlating genes are used to aggregate their expression and obtain the CytoTRACE score. The spot that has the lowest score will be the root.

**Pseudo-time-space algorithm.** The algorithm to model spatial trajectories based on gradient changes in transcriptional states for ST data has two main components: spot/cell location data (spatial) and gene expression data (which contains the pseudo-temporal changes). The PSTS algorithm does not require imaging information, but imaging features can be used optionally in the stSME pre-processing step. PSTS can otherwise also be applied to multiple types of data that are either with or without single-cell resolution, like sci-Space single-cell data (Figs. 3a, b and S7) or Visium spot-level data, respectively. PSTS is described in pseudo-code in **Algorithm** 1.

**Algorithm** 1. Pseudo-time-space for two clusters

```
1:  for  Every cluster M_i do
2:       FindSubclusters(M_i).
3:  Assign root r (specified cluster by user, automatically
    choose root spot).
4:  for  Every sub-cluster m_i do
5:       if  Length(m_i) < threshold then
6:            Remove(m_i) #(optional, to exclude small sub-
            clusters)

7:       FindCentroid(m_i)

8:  for  Every pair of sub-cluster m_x and m_y do
9:       CalculatePseudotimeDistance dPT(m_x, m_y)

10: Construct spatial-PAGA graph
11: Choose a cluster M = m_1, m_2, m_3, . . . , m_n.
12: Construct a graph D(M, E), E = e_1, e_2, ...e_n. # E edges,
    D directed graph
13: for  Every pair of connected nodes m_i and m_j do
14:      e_i = d_PTS(m_i, m_j) (Equation 6) by both d_PT (Equation
         4) and d_S (Equation 5).
15: AddDirection(D)
16: # For the global analysis:
17: Choose clusters U = u_1, u_2, u_3, . . . , u_n and V =
    v_1, v_2, v_3, . . . , v_n.
18: Construct a bipartite graph D(U, V, E), E = e_1, e_2, ...e_n
19: for  Every pair of connected nodes u_i and v_j do
20:      e_i = d_PTS(m_i, m_j) (Equation 6) by both d_PT (Equation
         4) and d_S (Equation 5).
21: AddDirection(D)
22: Add a pseudo-root r_pseudo that connected with
    u_1, u_2, u_3, . . . , u_n.
23: MinimumSpanningArborescence(D)
24: If > 2 nodes (clusters) needed to connect U to V, then find
    optimal path from U to V in the spatial-PAGA graph by
    minimizing the total physical distance of the path
25: Trim all edges outside of optimal trajectory connecting
    nodes U and V
```

**Pseudo-time-space values calculation.** PSTS starts with calculating pseudo-temporal values for each spatial spot. We added a spatial computational layer on top of the Diffusion Pseudotime (DPT) algorithm[48], taking into account spatial proximity to compute PSTS distances as described below. We ran a modified DPT for all spots of the tissue and, as part of the DPT algorithm, then applied a semi-automated approach to determine a root as described above for the targeted biological process (e.g. a non-invasive sub-cluster as the root of a cancer progression process). Changes in the DPT values reflect

pseudo-temporal changes in gene expression here across clusters. Thus, when analysing a spatially captured array, these modified DPT pseudo-temporal values that now take into account the spatial information can be defined as PSTS values, and we hence refer to them as such in our algorithm.

**Calculating pseudo-time-space distance ($d_{PTS}$).** Given two sub-clusters $u$ and $v$, to calculate the distance between gene expression profiles, we set PCA components as the feature vector $pu_i$ or $pv_i$, respectively, which represent the gene expression state of a spot/cell. Next, we calculate the cosine distance between all pairs of the feature vectors $pu_i$ and $pv_i$ to observe the gene expression distance between each pair of spots of $u$ and $v$. We then take the mean of those distances to get the gene expression distance of the two sub-clusters $u$ and $v$. As the gene expression change can represent the temporal information, we treat the gene expression distance here as the pseudo-temporal distance $d_{PT}$.

$$d_{PT(u,v)} = \sum_{i=1, j=1}^{n} \frac{1 - \frac{pu_i \cdot pv_j}{\|pu_i\|_2 \|pv_j\|_2}}{n} \qquad (1)$$

The spatial distance $d_{S(u,v)}$ between the two sub-clusters is calculated as:

$$d_{S(u,v)} = \sqrt{\sum_{i=1}^{n} (cu_i - cv_i)^2} \qquad (2)$$

where $cu_i$ and $cv_i$ are the coordinates of the centroids of sub-clusters $u$ and $v$. By combining the pseudotime distance with the spatial distance, we can now compute the pseudo-time-space distance $d_{PTS(u,v)}$ as follows:

$$d_{PTS(u,v)} = d_{PT(u,v)} \times \omega + d_{S(u,v)} \times (1 - \omega), with \, \omega \in [0,1] \qquad (3)$$

where $\omega$ is a weighting factor reflecting the balance between gene expression and physical distance (discussed below).

The maximum $PT$ score across all spots in each cluster/sub-cluster is used as a representative value for the given location because it reflects the difference in gene expression between this (sub-)cluster and that of others. We found that, even at the sub-cluster level, where spots are most similar, there is still a high variation of gene expression among all spots. For example, the distribution of $PT$ values in different sub-clusters is stochastic (Fig. S11d). Therefore, the single mean value or the summation of all individual spots is not sufficient to represent the transcriptional state(s) of sub-clusters, and the maximum value thus appears to be a better metric. A reasonable assumption for the transcriptional states of individual spots within a sub-cluster is that two spots that have a greater physical distance within a sub-cluster are more transcriptionally different than two more proximal spots[20–22]. Overall, $d_s$ reflects the relationship between two sub-clusters by calculating the physical distance (2), and $d_{PT}$ reflects the relationship between transcriptional profiles (1). An important parameter here is $\omega$ which, as mentioned earlier, represents the weights by which gene expression and spatial distance effects contribute to calculating the $d_{PTS}$ (3). If $\omega = 1$, then only gene expression is considered. Conversely, for $\omega = 0$ only physical distance is considered. Intermediate values of $\omega$ incorporate both gene expression and physical distance to different degrees, allowing the user to assess the relative contributions of these two measures in the graph optimisation step (Fig. S10). We developed a quantitative stLearn function to assess the effect of $\omega$ on the model result using the graph Laplacian distance as described later.

Applying the formulae above, we build an adjacency matrix from $d_{PTS}$ for input into the PSTS analysis method at both the local and global clustering levels.

**Spatial topology-preserving map construction.** Spatial-PAGA is the type of graph that we developed to reconstruct the spatial trajectory/trajectories. It is based upon PAGA[49], but can generate a topology-preserving map of spots with gene expression and spatial information. It also provides a preliminary general structure of the relationships in gene expression across clusters. With the clusters as nodes of the graph, only nodes with connected edges can reconstruct the spatial trajectories. Nodes are split into multiple nodes if they represent clusters with multiple sub-clusters. With the edges of the graph, we computed distance (between nodes) as $d_{PTS}$.

**Spatial trajectory reconstruction.** The main part of the PSTS algorithm aims to find how multiple clusters and/or spots are connected within a tissue (Figs. 1c, S1, and S8).

Given two sets of sub-clusters: $U = \{u_1, u_2, u_3, ..., u_n\}$ and $V = \{v_1, v_2, v_3, ..., v_n\}$ in two separate clusters $U$ and $V$, we can first order sub-clusters for each of the two clusters $U$ and $V$ by ranking the sub-clusters' minimum spot $pt$ values (e.g. a spot with the lowest $PT$ among all spots in the sub-cluster $v_1$). Note that $U$ and $V$ can be found as the clusters with the minimum and maximum of $PT$ values. On the assumption that the overall $PT$ order is from $U$ to $V$, we denote that the dynamic process shifts from $U$ to $V$.

We then build an adjacency matrix using $d_{PTS}$ where the dimensions of the matrix are the number of $u_i$ nodes in $U$ and $v_j$ nodes in $V$ from two sets of sub-clusters $U = \{u_1, u_2, u_3, ...u_n\}$ and $V = \{v_1, v_2, v_3, ...v_n\}$. This means that the values of the distance matrix are sets of $d_{PTS}$ between every two sub-clusters $u_i$ and $v_j$. We compressed each sub-cluster to become a node in the graph, and the distance between two nodes ($u_n -> v_n$) is $d_{PTS}$ (3).

From the adjacency matrix of $d_{PTS}$, we build the spatial-PAGA graph, which is a directed and bipartite graph comprising the initial trajectories (5) that capture the directions, based on the selected root, from sub-clusters of $U$ to sub-clusters of $V$. For example, a bipartite graph is constructed with $D(U, V, E), E = \{(u_1, v_1), (u_1, v_2), (u_1, v_3), ...$ and $(u_2, v_1), (u_2, v_2), (u_2, v_3), ...,$ and so on until $(u_n, v_n)\}$. From the fully connected directed spatial-PAGA graph, a pseudo-root is added to the graph to form an arborescence (a rooted, directed tree), which can be optimised by using a minimum directed spanning tree approach with Chu-Liu/Edmonds' algorithm[62]. This yields a weighted, directed graph $D(N, E)$ where $N$ is the set of nodes (sub-clusters), $E$ is the set of directed edges ($D(N, E)$ contain 'raw' trajectories), a node $r$ called root (assigned pseudo-root) in $V$, and $\omega$ is the weight of each edge in $E$ (calculated as $d_{PTS}$). From the 'raw' fully connected tree, we identified a directed spanning tree or spanning arborescence $A$ with a root at $r$ such that every node in $A$ has two edges (in and out, except for the tip of the branch, which has one edge). The optimisation process (to find optimum branching) is performed such that $A$ has a minimum weight, defined as the sum of all edge weights in $A$ as the cost function:

$$w(A) = \sum_{e \in A} w(e) \qquad (4)$$

where $e$ is the edge weight.

After finding the minimum directed spanning tree, we obtain the optimal graph, for example, $D(U, V, E), E = \{(u_1, v_1), (u_2, v_2), (u_2, v_3)\}$, (Fig. 1c) which represents the trajectory of each sub-cluster from the lower layer (i.e. lower PSTS values) to the higher layer. With this approach, one node can be the start node of multiple branches but the end node belongs to one branch only. Finally, we overlay the branches on the tissue image to allow for the visualisation of the spatial trajectories.

$$D_{global}(U, V, E), E = \{d_{PTSuv1}, d_{PTSuv2}, ... d_{PTSuvn}\} \qquad (5)$$

By letting $F$ be a set of $n-1$ edges extracted from $D$, we can determine the cheapest edge (an edge with the lowest weight) entering each node $v \neq r$ that always forms a path $v_0 \leftarrow v_1 \leftarrow \ldots \leftarrow v_n$, where each $v_i$ is an original node. Because of the set-up of the graph, we obtained $F$ as a directed acyclic graph for the simplest scenario (without any cycle in the graph) of this algorithm. In detail, for the initialisation step, $F$ is empty. At each step, the algorithm selects an arbitrary node $v \neq r$, which does not yet have an incoming edge in $F$, it then finds the cheapest edge $(u, v) \in E$ entering $v$, and adds $(u, v)$ to $F$. To find the cheapest edge entering a given node, the algorithm repeatedly executes minimum weighted edge extraction operations until the returned edge is not a self-loop in the current graph.

If there are clusters between $U$ and $V$, we offer an option to determine the optimal route connecting $U$ and $V$ in the spatial-PAGA graph. First, we generate all conceivable paths from $U$ to $V$ within the graph. Subsequently, we employ the minimum spanning tree (MST) algorithm (6) to identify the shortest path among the available options. Specifically, this algorithm computes the total weight of all edges, treating them as distances along each potential path (multiple source and target pairs are permitted if $U$ and $V$ possess multiple sub-clusters):

$$MST(U,V) = \min \sum_{U \to V \in S} d(U \to V) \tag{6}$$

where $S$ represents the set of edges of all possible paths, $d(U \to V)$ denotes the total weight of all edges connecting nodes $U$ and $V$, and min indicates the minimum value over all possible edges in the graph. From this, the path with the lowest edge weight value can then be selected.

**Optimisation of weighting parameter $\omega$ in $d_{PTS}$ calculations.** The weighting value $\omega$ is used to balance the spatial information and gene expression contribution to the $d_{PTS}$ for different biological samples and/or questions. We can quantitatively disentangle the contribution of spatial versus gene expression data here by comparing graphs where $\omega$ changes from 0 to 1, with two references graphs constructed that either do, or do not, use spatial information. Put differently, one can visualise how the graph changes and/or differs from the two reference graphs that use either just gene expression ($\omega = 1$) or spatial distance ($\omega = 0$) alone. This comparison is made possible by using graph adjacency matrices. In short, for each $\omega$ value, we compare the dissimilarity between the resulting graph ($G_\omega$) with its two references ($\omega = 1$, $G_1$ and $\omega = 0$, $G_0$). Laplacian distances are used for graph comparisons[63,64]. For example, if $A$ is an adjacency matrix of a graph $G$, to compute the distance between two graphs, we need to calculate the spectrum of each graph (an ordered set of eigenvalues of the graph's adjacency matrix). We defined matrices $A$ (adjacency), $H$ (diagonal) and $L$ (Laplacian) as:

$$A_{i,j} \stackrel{def}{=} \begin{cases} w_{i,j} & \text{if } i \sim j \text{ (if } i \text{ and } j \text{ are adjacent)} \\ 0 & \text{otherwise.} \end{cases} \tag{7}$$

A node in graph $G$ will have the degree $d_i \stackrel{deff}{=} \sum_{j \sim i} w_{i,j}$. The diagonal matrix of degrees is the degree matrix $H$, so $H_{i,i} = d_i$ and $H_{i,j} = 0$ for $i \neq j$. The combinatorial Laplacian matrix of $G$ is given by $L \stackrel{deff}{=} H - A$. To represent $G$, we therefore used two matrices $A$, $L$. The sorted sequence of eigenvalues means the spectrum of each matrix, represented as below, where the $k^{th}$ eigenvalue in descending order of the adjacency matrix is denoted as $\lambda_k^A$:

$$\lambda_1^A \geq \lambda_2^A \geq \ldots \geq \lambda_n^A \tag{8}$$

For the Laplacian (L) matrix, we have $\lambda_k^L$ and the $k$th eigenvalue, in ascending order

$$0 = \lambda_1^L \leq \lambda_2^L \leq \ldots \leq \lambda_n^L \tag{9}$$

The **L** matrix has at least one 0 eigenvalue, and the number of 0 eigenvalues is equal to the number of disjoint parts in the graph[63].

Letting $G$ and $G\prime$ be graphs that we want to measure, as in, the distance between them, we can compute Laplacian spectra $\lambda^L$ and $\lambda^{L\prime}$ for $G$ and $G\prime$, respectively. The Laplacian spectral distance between the two graphs is defined as:

$$d_L(G,G\prime) \stackrel{def}{=} \sqrt{\sum_{i=1}^{n} (\lambda_i^L - \lambda_i^{L\prime})^2} \tag{10}$$

We measure the graph dissimilarity between $G$ and $G\prime$ by calculating the absolute of value of the balance score between both the physical distance and gene expression-based graph (11).

$$\min_{\omega=0}^{1} |1 - (d_L(G_\omega,G_0)/d_L(G_\omega,G_1))| \tag{11}$$

The optimal $\omega$ is the one that balances the divergence from the two references, and its weighting value can be quantitatively determined through an $\omega$ sensitivity analysis where, as alluded to earlier, the contributions of spatial and gene expression components to corresponding PSTS trajectory graphs are assessed when $\omega$ changes from 0 to 1 (in 0.01 increments); this quantitative approach uses two reference graphs that use either only spatial information ($\omega = 0$), or gene expression data ($\omega = 1$; Fig. S10a). The point where the dissimilarity score is lowest optimally balances the individual contribution(s) of both the spatial and gene expression component towards the spatial trajectory being reconstructed. To also assess variation in the optimal $\omega$ parameter across different biological systems, we compared the determined optimal value for three independent ST-seq datasets (mouse TBI, mouse neurodevelopment and human breast cancer progression), finding that the optimal $\omega$ was relatively similar (0.46–0.51; Fig. S10a). To also illustrate how the optimal $\omega$ outperforms other choices, we show that gene expression bias ($\omega > 0.46$) can lead to non-specific trajectories, as exemplified in the DCIS-IDC case study (Fig. S10b) where a single DCIS (sub-cluster 6) now connects to either multiple or all IDC sub-clusters (Fig. S10b, right lower panels). This outcome goes against the findings of de Bruin et al.[65] who reported evidence for spatially-branched evolution of cancer clones, with the multiple branches being derived from different clones (supported in our dataset by the transcriptional relationships/diversity within and between DCIS-IDC pairs). With spatial bias, on the other hand (e.g., when using spatial distance alone ($\omega = 0$); Fig. S10b, left lower panel), we found a PSTS graph that appeared to accurately reflect the relationship(s) between DCIS and IDC clones (sub-clusters 6, 13, 16); this could be due to the edges from sub-clusters 6 and 16 in the initial graphs before optimisation (i.e., PSTS distance) already being stable, but the actual edge weights increased when $\omega$ changed from 0 to 0.46 (optimum). It is important to recognise here that spatial bias assumes nearby cells are more likely connected in the spatial progression process, which may lead to "over-fitting", that is, predicted branches wrongly connecting to the nearest cells; "gaps" or "missing nodes" within the spatial data of a thin 2D tissue section may pose an additional issue for inferring continuous trajectories. Indeed, as exemplified in Fig. S10c, a tissue section may contain a gap between connected clusters involved in cell movement/translocation in instances where the intermediate node is not within the same section but rather the third dimension (depth) of the tissue block, causing perspective bias. In stLearn's PSTS, we utilise the gene expression information to capture this missing structure of the graph because the gene expression data still contains the global transitional pattern and/or

relatedness of cells, even without the spatial connection (as observed in PCA/UMAP/Pseudotime latent space). Our optimisation step for $\omega$ therefore overcomes issues associated with both sources of bias, with the identified optimal value of $\omega$ (0.46 in this example; Fig. S10b) providing the right balance between the contribution of spatial and gene expression data, and a good fit to the actual spatial relationships that prevent trajectories from being misaligned with the biological process.

**Parameters setting.** Lastly, to run the PSTS analysis, there are two settings that need to be specified. The first are the tissue regions of interest within which to map the spatio-temporal processes. In our TBI case study, we chose four main anatomical regions (i.e. hypothalamus [node 4], thalamus [node 2], hippocampus [node 3] and penumbra regions of the damaged site [node 1]), the reason being that route 4-2-3-1 was identified through an unsupervised process as one of the shortest path from the terminal state (cluster) 4 to the initial state (node 1) (based on PSTS values) in the spatial-PAGA graph (see Fig. S6b and Supplementary Note 1.3). The second setting that needs to be specified is the predicted root cluster for a given biologically meaningful process that is expected to occur in and/or across the tissue of interest. As alluded to earlier, we adopt the CytoTRACE algorithm to automatically find the root cell/spot that has the lowest differentiation status for the chosen cluster. It should be noted that, depending on the biological question, the chosen root cluster could represent either the beginning or endpoint of any such process. stLearn therefore includes a reverse function that allows the root to be converted to the tail. Since our interest for the TBI case study was to map and visualise microglial activation states, we treated the damaged site as an endpoint or, put differently, as the site at which activation is most advanced while regions in the trajectory that are more distant from the damaged site would represent no or earlier stages of activation. Lastly, to account for the fact that a given ST technology may not necessarily be at single-cell resolution, we have included an optional third parameter where the user can specify a gene set for a given cell type (e.g. from RNA-seq studies) to better target the analysis to relevant cell types only.

**Detection of trajectory-based transition markers.** As a part of the PSTS analysis, stLearn allows users to find genes that likely drive the spatial trajectory. For a chosen clade $B_i$ (i.e. a branching trajectory path) and for a set of genes $E$ with gene expression values $e_1, e_2, \ldots e_n$, genes that are differentially expressed between the start and end nodes in a trajectory pathway are further tested for significant correlation using Spearman's rank order test between the gene expression value and pseudotime (adjusted $P$-value < 0.05). Spearman's rank correlation coefficients for $r_j$ will be high if the gene expression $e_j$ is linearly correlated with $PT_j$ in clade $B_i$. After multiple testing correction, significant genes are predicted to represent trajectory-driving genes. Transition genes with absolute Spearman's rank correlation coefficients that are higher than a specified threshold (0.3 by default) are classified as upregulated (if > 0.3), or downregulated (if < −0.3).

**Gene set enrichment analyses.** Functional enrichment analyses were performed for the top 100 up- and downregulated transition genes identified from the trajectory-based differential expression analysis described above; the Wiki Pathway Mouse 2019 database was used as the reference, and implemented using gseapy[66] and Enrichr[67]; an adjusted $P$-value < 0.05 was set as the threshold to filter significant pathways.

**Experimental validation of PSTS**

**Establishing an experimental model to visualise microglia activation in TBI mice.** Three- to four-week-old CX$_3$CR1$^{creERT2}$ x tdTomato$^{flox/flox}$ mice were orally gavaged with tamoxifen (12.5mg/g body weight) once daily for five days. Mice were then left to rest for a six-week period, which allows for the peripheral turnover of shorter-lived CX$_3$CR1-expressing leucocytes in the circulation and thus only leaves self-renewing CNS-resident tdTomato$^{pos}$ microglia. These mice were subjected to controlled cortical impact or sham surgery (craniotomy only), as described earlier, and then sacrificed at the specified time points.

**Tissue processing, staining and imaging.** Mice were euthanised using sodium pentobarbitone (1.6 mg/g body weight; i.p. injection), followed by transcardial perfusion with phosphate-buffered saline (PBS) and 4% formalin. Brains were dissected, post-fixed for 24 h at 4 °C, and then stored in PBS with 0.01% sodium azide until further processing. For sectioning, tissues were cryoprotected in 30% sucrose with 0.05% sodium azide for 2 days at 4 °C, after which serial brain sections (40 μm; 1 in 6 series) were cut using a sliding microtome (Leica).

Free-floating sections were washed in PBS, stained with DAPI (1: 1000; Sigma-Aldrich), and then mounted and coverslipped using Vectashield H-100 medium (Vector Labs). Nucleated tdTomato$^{pos}$ microglia were counted in four anatomically defined brain regions (i.e., nodes 1–4 of the spatial trajectory) for each mouse. Cell counts were performed live at ×400 magnification using StereoInvestigator software (MicroBrightfield Bioscience) on an inverted AxioImager Z2 microscope (Zeiss) with an ORCA-R2 digital charge-coupled device camera (Hamamatsu) and 40× 0.75 $NA$/0.71$mm$ WD objective. For each condition and time point, cells were counted within a field of view and then normalised to the area of the field of view (0.0755$mm^2$). Representative confocal images of the analysed brain regions were also acquired using an inverted Diskovery spinning disk confocal microscope (Spectral Applied Research) with 0.3$μm$ Z-stacks using a CFI Apo Lamda 60x Oil / N.A. 1.4 / W.D. 0.13$mm$ objective and Zyla sCMOS camera. After pre-processing the images in Fiji, we manually segmented all microglia in these images and then measured their cell body size in pixel × pixel units. Cell counts and cell body sizes were compared both over time and across tissue regions. The Kolmogorov-Smirnov (KS) test was used to test for distribution differences in cell body size and density through time and/or space.

**Comparing spatial trajectory reconstruction with non-spatial scRNA-seq equivalent methods.** As there are no spatial trajectory methods available to suitably reconstruct branching processes within the tissue context, we could only assess the performance of our PSTS algorithm against trajectory inference methods originally developed for scRNA-seq[33], comparing the variation (deterministic nature) of the pseudotime values between spatial spots within a tissue. Higher spatial variation suggests a noisier spatio-temporal pattern of the trajectory here. We used the variogram model as a quality metric to assess the performance of PSTS, Slingshot[29] and Monocle3[30] (Fig. 2i, j and Supplementary Note 1.2). In spatial statistics, a variogram describes the spatial continuity of the data[68,69], which are pseudotime values in our case and where Variogram eigenvalues reflect the spatial variation of PSTS/pseudotime values in the dataset. To calculate the variogram plot as in Fig. 2i, we used a semi-variance estimator, also called the Matheron estimator ($\gamma(h)$), and defined as:

$$\gamma(h) = \frac{1}{2N(h)} * \sum_{i=1}^{N(h)} (x)^2 \qquad (12)$$

with:

$$x = Z(x_i) - Z(x_{i+h}) \qquad (13)$$

where $Z(x_i)$ is the PSTS/pseudotime value at i-th location $x_i$, $h$ is the distance lag (Euclidean distance between two data points, based on spatial coordinates), and $N(h)$ is the number of point pairs at that lag. In this case, the distance between lags is 167 pixels. A lower semi-variance value, as computed by the Matheron estimator, reflects the

higher PSTS/pseudotime continuity, that is, a better spatial trajectory.

## Spatial trajectory inference by PSTS from multiple datasets

We developed a workflow for integrating data from different tissue samples (Fig. S13a). In brief, and using the Visium human breast cancer dataset (Block A, sections 1 and 2) as an example, we designated section 1 as the reference and section 2 as the moving sample (Fig. S13b). We then utilised a geometric transformation method to register the spatial coordinates of the moving sample to match those of the reference sample. Specifically, to address the shifting issue between the two samples, we performed a linear transformation using a matrix of [[1 0 0], [0 1 1000], [0 0 1]]. By applying this transformation to each point in the moving sample, we translated its coordinates to align with those of the reference sample. As a result, the DCIS and IDC clusters were well-matched at the (sub-)cluster level between the two sections (Fig. S13b). Users have the option to implement other image registration methods as preferred (e.g., PASTE[70]); for batch correction of gene expression information, we recommend using popular methods such as Harmony[71]. Once the registration and batch correction steps have been completed, PSTS can now applied as shown in Fig. S13c, where we inferred results in the same way as with single-sample PSTS analysis (Fig. S13c). If a z-dimension is present, users need to set the radius parameter to cover the 3D distance to define neighbouring spots/cells in order to calculate PSTS values. For instances where image registration is not possible, each sample can be subjected to independent PSTS analysis, following which shared driver genes can be identified to annotate the common trajectory between sections in an unbiased manner (see "Results" and Fig. S13d–h).

## stLearn cell–cell interaction analysis

Most of the current cell–cell interaction (CCI) inference methods do not take into account the fact that cellular interactions operate the strongest within 0 to 200 μm distance[43,44]. Consequently, these methods face an inherent issue of detecting false positive interactions between cells that are located outside this biological distance limit[72]. We indeed observed this common phenomenon when applying existing CCI methods to spatial data (Fig. 5e, f). In this work, we developed stLearn SCTP to improve prediction accuracy by utilising the spatial context of cells and their entire ligand-receptor (LR) expression repertoire. We developed two test categories, one for testing neighbourhoods with significant enrichment of LR co-expression (neighbourhood LR analysis - to find spatial locations and significant LR pairs used for interactions), and a second for finding cell type combinations with significantly greater interactions than other cell types across the tissue (cell type-specific CCI analysis).

First, we define a neighbourhood for a given spot as the set of spots within a defined spatial distance of that spot. A distance of 0 indicates 'within-spot' mode, whereby the neighbourhood of a given spot only includes itself. This mode is useful in ST-seq data such as Visium, or in binned data of high-resolution technology like Slide-seq[15], where each spatial spot/bin may contain multiple cells operating within less than 200 μm. The 'within-spot' mode is thus suitable for studying juxtacrine, autocrine and paracrine interactions. 'Between-spot' mode considers adjacent neighbours of a spot within a given distance as the neighbourhood. For 'between-spot' mode (default), the distance is automatically calculated as twice the spot diameter in Visium data, or a bin in other types of data. stLearn's SCTP module for CCI analysis is flexible, however, in that the distance can be specified by the user, following which spot/bin/cell neighbourhoods are determined using both the spatial coordinates and distance parameter as the input into the cKDTree in SciPy[73].

*Location-specific LR calculation*: once neighbourhood spots/bins/cells are defined, LR co-expression ($LR_{score}$) for each spot is calculated

as per Eq. (14):

$$LR_{score} = \frac{1}{2}\left(mean(Expr_{L,S|N} \times [Expr_{R,S} > 0]) + mean(Expr_{R,S|N} \times [Expr_{L,S} > 0])\right) \quad (14)$$

where $Expr_{L,S|N}$ is the expression of the ligand in either the spot or the neighbourhood spots of the spot $S$ (denoted as $S|N$), and $Expr_{R,S} > 0$ is a conditional value of either 0 (if receptor R not expressed) or 1 (if receptor R expressed) in spot S. The terms $Expr_{R,S|N}$ and $Expr_{L,S} > 0$ are equivalent to the aforementioned but in reference to the receptor (R) and the ligand (L), respectively. In the case of within-spot mode, the $LR_{score}$ reduces to Eq. (15):

$$LR_{score} = \frac{Expr_{L,S} \times [Expr_{R,S}>0] + Expr_{R,S|N} \times [Expr_{L,S}>0]}{2} \quad (15)$$

The $LR_{score}$ can be further adjusted to include cell type diversity, which is often positively correlated with increased likelihood of cell–cell interactions[13,23,74]. The positive correlation between cell type diversity and cell–cell interaction activities is indeed well supported. An immune social network model demonstrated that context-dependent paracrine responses, which were quantified as cell–cell interaction connections, were correlated with the number of sender and receiver cell types[23]. A weighted-directed-multi-hyperedge network model found that the underlying structure of a real cell-to-cell communication network was made up of hyperedges where ligand-receptor pairs connect multiple cell types[24]. However, as the relationship between cell type diversity and cell–cell interaction may be dependent on biological contexts, stLearn SCTP implements cell type diversity as an optional parameter. To acquire cell type information, spot cell type scores between 0 and 1 can be derived from reference scRNA-seq datasets using provided python wrapper functions for tools such as RCTD[75] or Seurat v3 label transfer[76]. Different cell types are then counted per-spot when the cell score is greater than a given threshold ($C$; default 0.2) in either its neighbours (between-spot mode), or within itself (within-spot mode) as in Eq. (16):

$$HET_{spot} = \sum_{unique\_celltypes} P_{celltypes|S} > C \quad (16)$$

The $LR_{score}$ can then be adjusted to $LR'_{score}$ to allow for prioritisation of interactions in the areas of high cellular heterogeneity as in Eq. (17):

$$LR'_{score} = LR_{score} \times HET_{spot} \quad (17)$$

Of note, stLearn SCTP implements the Numba framework and parallel processing to improve the computation speed, enabling the fast calculation of $LR_{score}$, $HET_{spot}$, and $LR'_{score}$ for all spots, LR pairs and random gene-gene pairs.

*LR significance testing*: we introduced a robust statistical method to test LR interactions, avoiding biases towards abundant LR pairs and random co-expression of non-interacting pairs of genes across neighbour spots/cells. To do so, we established a random background of LR scores for $2 \times \sqrt{k}$ non-interacting genes, these being genes that are not in the LR database (NATMI[24] by default) but within the same expression ranges compared to each of the ligand and receptor genes that constitutes the LR pair for testing. To select random candidate genes in the same expression range of a given query gene (query gene as either the ligand or receptor in the LR pair), we find genes that minimise the Canberra distance between $n$ quantiles of gene expression across all spots/cells, i.e. to select $2 \times \sqrt{k}$ genes with the lowest ($g_{dist}$) as in Eq. (18)

(illustrated in Fig. 4a):

$$g_{dist} = \sum_{i=1}^{n} \frac{|q_i - g_i|}{|q_i| + |g_i|} \tag{18}$$

where $n$ is the total number of quantiles (default 10), $q_i$ is the value of quantile $i$ for the query gene receptor or ligand, and $g_i$ is the value of quantile $i$ for a candidate random gene. A quantile $i$ for a gene is defined by the expression values of that gene across all spatial measurements of a dataset (e.g. all spots in a Visium dataset). Due to the high proportion of zeros commonly observed in ST-seq data, the quantiles used to select candidate random genes are oversampled toward the higher end of the gene expression distribution to ensure quantiles do not have all 0 values for sparsely expressed genes; by default quantiles 50, 75, 85, 90, 97, 99, 99.5, 99.75, 99.99 and $100th$ are selected. These random genes, selected to represent the ligand expression and the receptor expression, are then randomly paired to generate non-interacting gene-gene pair with equivalent expression levels to the LR pair (Fig. 4a). We then calculate the $LR_{score}$ (or optional $LR'_{score}$) for each random pair to create the background distribution per spot and LR pair (Figs. 4a and S17). As a result, for each LR pair, we sampled $n \times kLR_{score}$ values ($n$ is the number of spots/ bins/cells in the dataset, by default $k = 1000$).

With the background signal established, we calculate $P$-values for each spot and LR pair ($p_{s,LR}$) as the proportion of the background scores ($b_i$) across $k$ random pairs that had a score greater than the $LR_{score}$ (Benjamini/Hochberg correction for multiple testing). To assess whether a robust number of random pairs has been obtained in a dataset-specific manner, stLearn SCTP implements diagnostic plots that automatically examine the relationship between the LR expression level and abundance with the LR rank based on the number of significant spots (Figs. S14i and S15d–f). These diagnostic plots show no correlation between LR rank and LR expression if an adequate background has been generated.

*Cell type-specific interaction analysis*: in addition to testing for significant interactions by LR pairs at specific spatial locations, as described above, we also introduced significance testing for cell type-specific interactions, where we can test if a pair of cell types interact using a given pair of LR. This test accounts for the fact that more than one cell type may be present at a given spot, but that not all of these cell types are involved in the interactions within one spot or between two neighbouring spots.

The cell type interaction analysis uses the significant spot/bin/ cell outputs from the spot LR analysis above (Fig. 4a). For each LR pair and spot, we then calculate the count matrix $CCI_{LR}$ of shape $n_c \times n_c$, where $n_c$ is the number of all predicted cell types. Each row in $CCI_{LR}$ corresponds to the signal emitting cell types (ligand expressing; sender), and each column to the signal detecting cell types (receptor expressing; receiver). For a given cell type $x$ and cell type $y$, the row of $x$ and column of $y$ in $CCI_{LR}$ is the count of the receptor-expressing neighbour spots that contain cell type $x$ for significant spots that contain cell type $y$ and express the ligand. This also includes counts for the reverse situation, where the significant spot expresses the receptor and contains cell type $y$ while the neighbour expresses the ligand and contains cell type $x$. This counting is detailed mathematically in Eq. (19):

$$
\begin{aligned}
CCI_{x \to y|LR} = &\sum_{s}^{S} \sum_{n}^{N} [Expr_{R,n} > 0 \land celltype_n == y] \land \\
&[Expr_{L,s} > 0 \land celltype_s == x] + \\
&\sum_{s}^{S} \sum_{n}^{N} [Expr_{L,n} > 0 \land celltype_n == x] \land \\
&[Expr_{R,s} > 0 \land celltype_s == y]
\end{aligned} \tag{19}
$$

Here, $CCI_{x \to y|LR}$ is the count of cell type $x$ signalling to cell type $y$ via a given LR pair; $S$ is the total number of significant spots for the LR pair; $N$ is the total number of neighbours for spot $s$; $Expr_{R,n} > 0$ is 1 if the receptor is expressed in the neighbouring spot $n$ and 0 otherwise; while $celltype_n == y$ is 1 if spot $n$ contains cell type $y$ and 0 otherwise. $Expr_{L,s}$ and $celltype_s == x$ are equivalent terms that refer to ligand expression L in the significant spot $s$ that contain cell type $x$. In 'discrete' mode, spots will have a discrete cell type label; while in 'mixture' mode a spot is considered to have a given cell type if the cell type score in the spot exceeds a given threshold ($C$; default 0.2). In practice, mixture mode results in a greater number of possible interaction counts since a single spot/bin is considered to represent multiple cell types simultaneously (as in Visium data). For example, a neighbouring spot containing 3 cell types would be counted a total of 3 times if the neighbour spot meets the interaction criteria explained above.

We found that the cell information permutation process detailed below takes into account cell type background frequency to effectively call significant interactions, as there was no observable correlation between the number of significant interactions for a cell type and cell type background frequency (Fig. S15c).

After calculating the $CCI_{LR}$ count matrix for each LR pair, we then permute the cell type information associated with each spot $V$ times and re-calculate $CCI_{LR}$ with the permuted cell type information. In 'mixture mode', the scores of each cell type across spots are randomised independently of one another, effectively permuting cell type colocalisation combinations while maintaining the frequency of each cell type in the sample. In 'discrete' mode, the cell type labels of each spot are simply randomised. This randomisation process effectively counts the colocalisation of cell types with significant LR neighbourhoods while taking into account the background frequency of those cell types in the sample. $P$-values are then calculated as in Eq. (20):

$$p_{x \to y|LR} = \frac{\sum_{v=1}^{V} (b_v > CCI_{x \to y|LR})}{V} \tag{20}$$

where $b_v$ is the count of interactions between cell type x and y via the LR pair for permutation v; $b_v > CCI_{x \to y|LR}$ is 1 if the background interaction count for permutation $v$ is higher than the observed interaction count $CCI_{x \to y|LR}$, and 0 otherwise. This test is directional between cell type $x$ and cell type $y$.

As with the LR analysis, the number of permutations chosen can affect the results. To indicate whether a robust number of permutations has been obtained in a dataset specific manner, we provide a diagnostic plot that examines the relationship between cell type background frequency and number of significant interactions. This diagnostic plot shows no correlation between the cell type frequency and number of significant interactions if an adequate background has been generated, as exemplified from our analysis of the Visium breast cancer data (Fig. S15).

Optionally, instead of empirical p-values, a negative binomial test based on the null distribution of $LR_{score}$ can be used to find predicted interactions. As described above, $LR_{score}$ is calculated based on the linear combination of LR gene pair expression in neighbour spots (also the case when cell type heterogeneity is multiplied with the $LR_{score}$ to give $LR'$). For each LR, we fit a negative binomial distribution to the whole set of $LR_{score}$ values from all spots and all $k$ random gene pairs, using the maximum likelihood estimation procedure to estimate parameters mean $mu$, and heterogeneity $alpha$ (modelling variance as $mu + alpha \times mu^2$). The cumulative distribution function of the negative binomial distribution ($NB_{cdf}$) was used to check whether the observed $LR_{score}$ is just by chance or not. The probabilities are then adjusted by multiplying with the number of tests (i.e. Bonferroni

correction for the tests across the total number of spots in the tissue):

$$CCI - NB_{spot} = -Log_{10}\Big[1 - NB\_cdf\Big[CCI - merged_{LR,spot}, NB\_fit$$
$$[CCI - merged_{LR_i,spot}]\Big] * N_{spot}\Big], (i = 1, .., n\_pairs)$$

where $NB\_cdf$ is the cumulative distribution function of null distribution based on CCI scores of 1000 (default) random protein-protein pairs.

The SCTP algorithm is summarised in pseudo-code in **Algorithm** 2.

### Algorithm 2. Cell–cell interaction

```
 1: # Calculate LR_scores for each spot
 2: for Every spot/grid/cell S_i in the tissue do
 3:     Use spatial coordinates to define neighbours
 4:     # Calculate Ligand-Receptor score (LR_i)
 5:     if between-spot/grid mode then
 6:         for Every neighbouring spot/grid S_ij do
 7:             M_Receptori = Expr_{Receptor,S_{i,j}} ×
                     (Expr_{Ligand,S_i} > 0)
 8:             M_Ligandi = Expr_{Ligand,S_{i,j}} ×
                     (Expr_{Receptor,S_i} > 0)
 9:             M_LR+ = (M_Receptori + M_Ligandi) / 2
10:         LR_i = M_LR/(Count(S_ij)) (Equation 14)
11:     if within-spot/grid mode then
12:         LR_i = LR_score = 1/2 (mean(Expr_{L,S|N} ×
                [Expr_{R,S} > 0])
                + mean(Expr_{R,S|N} × [Expr_{L,S} > 0]) (Equation 15)
13:     # Optional: adjust LR_i by HET_i
14:     LR'_i = HET_i × LR_i
15: # Run LR permutation test
16: for Every LR do
17:     ligand_quantiles = compute n quantiles of ligand
            expression
18:     receptor_quantiles = compute n quantiles of receptor
            expression
19:     for each non-LR gene do
20:         gene_quantiles = compute n quantiles of gene
                expression
21:         ligand_dist = compute gene_quantiles Canberra
                distance from ligand_quantiles (Equation 18)
22:         receptor_dist = compute gene_quantiles Canberra
                distance from receptor_quantiles (Equation 18)
23:     ligand_random = select 2 × √k genes with lowest
            ligand_dist
24:     receptor_random = select 2 × √k genes with lowest
            receptor_dist
25:     for j in range k do
26:         random_pair = list(randomly select from
                ligand_random,  randomly  select  from
                receptor_random)
27:         for Every spot/grid/cell S_i in tissue with LR_i > 0
                do
28:             score_randomij = compute LR_i as above using
                    random_pair
29: for Every spot/grid/cell S_i in tissue do
30:     count_LRi = Count(LR_i > 0 for each LR)
31:     for Each LR with LR_i > 0 do
32:         p_i = Count(score_randomi >= LR_i) / count_LRi (equivalent
                Equation 20)
33:     compute corrected p-values for multiple testing
34: Optional: cell type information permutation test to call
        significant celltype-celltype interactions (Equation 20)
```

Lastly, we also provide an optional cell binning step, which can be applied to very large spatial single-cell datasets (Figs. 4e–g and S14). Here, the tissue space is segmented into grids according to a user-defined grid size prior to analysis with the stLearn SCTP pipeline described above (Fig. S14c). This allows for stLearn's CCI analysis to be scaled to datasets with potentially millions of cells (as the computation limit is only dependent on the number of bins).

**Functional enrichment analysis for significant CCI hotspots.** We used the Wilcoxon rank sum test to detect differentially expressed genes using Seurat[76], comparing the CCI hotspots with all the remaining spots. The top 100 most significant differentially expressed genes of CCI hotspots were used as input for Gene Ontology (GO) analysis by g:Profiler[77] to test for GO enrichment in the Biological Process category. We then plotted the term size, overlap of genes and adjusted P-values for the top 15 most significant gene ontology terms (Fig. S15g).

### Validation of cell–cell interaction

**mRNA in situ hybridisation of cancer tissue sections.** RNAscope HiPlex (ACD Cat. No. 324110) was performed on fresh-frozen BCC tissue sections to detect ligand-receptor interactions for *IL34* and *CSF1R*. The target probes were designed by the ACD RNAscope Team, and the assay itself was performed as per the manufacturer's instructions. Briefly, the section was fixed with freshly made 4% PFA for an hour, followed by dehydration and digestion processes. The slide was then hybridised with probes against *IL34* and *CSF1R* mRNA. The negative control slide was stained with RNAscope HiPlex 12 Negative Control Probe, which targets bacterial housekeeping genes. RNAscope Hiplex Amp 1, 2 and 3 reagents were subsequently added to amplify the signal from hybridised probes, with washing steps performed between incubation steps with each Amp reagent. Sections were then stained with RNAscope HiPlex Fluor T1-T4 reagent, followed by nuclear counterstain using DAPI. Finally, the slides were coverslipped using ProLong Gold Antifade Mountant (Fisher Scientific) for imaging. Sections were scanned using an Axio Z1 slide scanner (Zeiss) at 40x magnification with a 1.5 μm Z-stack interval. Imaging was performed using different filters including DAPI for nuclei, *Cy5* for *IL34* and *Cy7* for *CSF1R*; TIFF images were generated using ZEN software (version 3.2) for further analysis of ligand - receptor interaction using customised scripts[78].

### stSME spatial imputation

ST provides transcriptome-wide gene expression profiles with additional spatial location. For many technologies, tissue morphology information from H&E tissue images can also be obtained for either the same or an adjacent tissue section. We developed the stSME imputation method to incorporate these two additional data types (that is, spatial location and tissue morphology) to adjust gene expression values between spots. Optionally, we provide stLearn's image processing functionalities and a neural network model to utilise the H&E image spot data.

**Matrix D - spatial location.** First, we calculate a spatial distance matrix **D** to utilise spatial location information. Different to spatial smoothing approaches, which use **D** to select neighbouring spot pairs for gene expression adjustment, here we integrate **D** with additional measures to select reference spots, as described later. The role of **D** is based on the assumption that spots which are close in physical distance are also likely to have similar gene expression profiles and cell types[20–22]. Physical distance $PD_{ij}$ is defined as the centre-to-centre Euclidean distance of any two spots $S_i$ and $S_j$ (21). The pairwise physical distance matrix $D$ is calculated for all pairwise spot combinations, including for near and distant spots in one tissue slide. For each spot $S_i$, the closest three layers of spots (i.e. spots that are within a radius of 3x $PD_{(S_i, S_{i\_adjacent})}$

from the spot $S_i$, calculated as below) are selected:

$$PD_{i,j} = \sqrt{(x_i - x_j)^2 + (y_i - y_j)^2} \tag{21}$$

where x and y denote the pixel coordinates of the centre of each spot/cell/grid. Matrix **D** is a symmetric square matrix with element $D_{i,j}$ between each pair of $n$ spots/cells/grids across the entire dataset.

**Matrix G - gene expression correlation.** Additional gene expression profile checks are implemented to avoid selecting neighbouring spots that are not of the same cell type. We first apply log transformation of raw gene expression UMI count data and then embed it into a PCA space. Gene expression correlation $GD_{ij}$ is calculated by Pearson correlation of spot $S_i$ and spot $S_j$ in low dimensional space (top 50 PCs) (22).

After initial spot selection by matrix $D$, only the three spots with highest correlations with spot $S_i$ are selected based on Matrix $G$ for downstream imputation (22):

$$GD_{i,j} = r(G_i, G_j) \tag{22}$$

where $G_i$ and $G_j$ are low dimensional PCA embedding of gene expression for spots $S_i$ and $S_j$ (10 PCs by default). Similarly to the matrix **D**, the symmetric square matrix **G** was calculated for every pair of $n$ spots.

**Matrix I - morphological similarity.** We measured the morphological similarity between paired spots $S_i$ and $S_j$ by the distance calculated based on two numeric vectors extracted from their corresponding spot images, i.e. the two H&E tiles that cover these two spots/cells/grid (tissue regions of interest). The high-level features of these spot images are extracted by ResNet50[79], which is a well-established convolutional neural network (CNN) model widely used for image classification in computer vision. We used a transfer learning strategy, where the weights in ResNet50 were pre-trained using the ImageNet[80] dataset (millions of images) and the final classification layer extracted. This method leverages the generalised ability of the pre-trained model in a large imaging dataset to extract numeric features from a new image[81]. As a result, the model can convert an image into a 2048-dimensional latent vector, and combining the latent vectors from all spots thus forms a latent space with 2048 dimensions. We further applied PCA to extract the first 50 PCs as the latent features to represent the spot morphology. The morphological distance $MD$ of a centre spot $S_i$ and its neighbour $S_j$ can be calculated by cosine distance (other options such as Euclidean or Pearson distances are also available in stLearn), computed as in Eq. (23):

$$1 - MD_{i,j} = 1 - \frac{M_i \cdot M_j}{\| M_i \| \| M_j \|}, \tag{23}$$

where $M_i$ and $M_j$ represent the morphological latent feature vectors for spots $S_i$ and $S_j$. $MD(S_i, S_j)$ is then scaled from the range $[-1,1]$ to $[0,1]$, where 0 means the most different and 1 means identical, with regard to morphological similarity.

**stSME weighting matrix.** The stSME matrix integrates information from physical distance (matrix D), gene expression correlation (matrix G) and morphological similarity (matrix I) to enable accurate imputation of ST data without losing spatial resolution. The units of measurements are cancelled out in the morphological similarity and gene correlation matrices. The physical distance is measured in pixel units, and can also be converted to micrometre unit, but in either case, the unit choice does not affect the result because the matrix is used for determining local spots. Common smoothing methods non-selectively use neighbour spots for imputation, thereby loosing localised variation. stLearn does not suffer from this limitation as the stSME matrix

allows users to simultaneously select spots with the nearest D, highest G and highest I as the reference spots for imputation. With these selection criteria combined, there can be a very high level of confidence that reference spots likely comprise the same cell types and cell states. Each element $W_{i,j}$ in *stSME* weighting matrix is the multiplication product of $GD_{i,j}$ and $MD_{i,j}$ from matrix $G$ and $I$, respectively, where spots $S_i$ and $S_j$ selected from matrix $D$ (24). Three spots with highest weights are then identified and their stSME weights normalised by total weights as the final weighting parameter for stSME imputation (25):

$$stSME = [W_{i,j}] = [GD_{i,j} \cdot MD_{i,j}] \tag{24}$$

$$W'_{i,j} = \frac{W_{i,j}}{\sum_{j=1}^{n} W_{i,j}} \tag{25}$$

where $n$ is the number of reference spots selected. $W'_{i,j}$ are the weights used for adjusting the expression value of spot $S_i$, which add up to 1 (Fig. S19c).

Incorporating spatial location, gene expression and morphological similarity, stSME normalises the gene expression of each spot $S_i$ by:

$$GE'_i = \tfrac{1}{2} GE_i$$
$$+ \tfrac{1}{2} \left\{ \sum_{j=1}^{n} (W'_{i,j} \cdot GE_j) | S_j \text{ selected by matrix } D \right\} \tag{26}$$

where $GE'_i$ represents stSME-normalised gene expression for a centre spot $S_i$. $GE_i$ and $GE_j$ represent raw gene expression for spot $S_i$ and its $n$ selected neighbour spots $S_j$, respectively.

**Algorithm 3.** stSME imputation

```
1: Calculate pairwise Euclidean Distance matrix D based on
   spots physical distance
2: Calculate pairwise Pearson Correlation matrix G based on
   spots' gene expression profiles
3: Extract spots' (S) images features (F) using pre-trained
   ResNet50
4: Calculate pairwise Cosine Similarly matrix I based on spots
   images features (F)
5: for Every spots Sᵢ do
6:     Select neighbouring spots from matrix D
7:     Calculate weighting parameter Wᵢ,ⱼ = GDᵢ,ⱼ · MDᵢ,ⱼ
8:     Normalise weighting parameter W'ᵢ,ⱼ = Wᵢ,ⱼ / Σⱼ Wᵢ,ⱼ
9:     Normalised gene expression GE'ᵢ for spots Sᵢ = ½ GEᵢ
       + ½ Σ ( W'ᵢ,ⱼ · gene expression GEᵢ,ⱼ of spots Sᵢ,ⱼ )
```

### Leave-out simulation for evaluating stSME imputation

Two simulation approaches were developed to evaluate the performance of the stSME imputation method; both used the raw gene expression matrix as the ground truth.

In the first approach, 20% of all data points were randomly selected and replaced with a "0" to simulate missing values. stSME imputation was then applied to recover these missing values, followed by Louvain clustering. Performance of the stSME imputation algorithm was evaluated by using the adjusted Rand index (ARI), assessing clustering results against the ground truth (i.e. original clustering obtained from the full data; Fig. 6d, e).

In the second approach we iterated through every spot on the tissue slide, treating each as being without gene expression values to simulate missing spot scenarios and then using the original gene expression of the nearby spots as a reference. The stSME method was then applied again to independently impute the top 2000 highly

variable genes (HVGs) across two replicate sections of the Visium human breast cancer dataset (10x Genomics; Block A, sections 1 and 2). After all spots were imputed, the spatial autocorrelation of the imputed gene expression matrix was calculated against the original gene expression matrix for each HVG to evaluate the performance of the stSME imputation, both within and across sections (Fig. 6f).

### Global and local spatial clustering - SMEclust

scRNA-seq data has led to the development of numerous cell type identification methods using data-driven clustering[82]. These clustering methods, such as those in the popular Scanpy[83] and Seurat[84] pipelines, only use gene expression information and it often remains challenging to decide on the resolution and number of clusters[8]. stLearn's spatial clustering (SMEclust) uses stSME-normalised data and Louvain clustering to first find clusters at the global level. We also introduce a feature in SMEclust to help decide the clustering resolution by using spatial information. Sub-clustering or local clustering is performed when one cluster consists of spots located in multiple different parts of the tissue, for example DCIS sub-clusters in breast cancer tissue (Fig. S8c–d). A usage example of broad/global clusters $U$ and $V$ and specific/local sub-clusters: $\{u_1, u_2\} \in U$ and $\{v_1, v_2, v_3\} \in V$ is shown in Fig. 1c.

### Implementation of software

Here we briefly describe the technical details for stLearn software implementation, which comprehensively incorporates three key analysis algorithms. Detailed tutorials and documentation are available on the stLearn website and the GitHub page (DOI:10.5281/zenodo.8251742). stLearn's core is constructed based on TensorFlow[85] and Pillow by Alex Clark for deep learning and image processing modules; SciPy[73] for the spatial analysis module; Scanpy[83] for the gene expression analyses modules; NetworkX[86] and Matplotlib[87] for network analysis and visualisation; and AnnData[83] for the main object to store and process data. As stLearn is a Python-based software, it is compatible with machine learning or deep learning packages like scikit-learn[88] or TensorFlow[85]. Users can also utilise functions from popular image processing packages, such as OpenCV[89] or Pillow for further analysis of tissue images, for example to extract a wide range of image features; these include stLearn's functions for segmentation of cell nuclei and other handcraft features, as shown in Fig. S20a). stLearn uses annData object and is compatible to a large range of other R/Python software that use the common SingleCellExperiment object, thus allowing convenient integration of analysis pipelines.

### Interactive stLearn web application

To extend usability to the broader scientific community, we have created a publicly available interactive version of stLearn called i-stLearn, based on Bokeh, an Interactive Data Visualisation framework (Fig. S25)[90]. Currently, the web application supports pre-processing of data, gene plots (including for LR pairs), clustering plots, the pseudo-time-space method and its visualisation, and cell–cell interaction analysis. Users can run the above analyses without the need for coding. The interactive functions can also run in the Jupyter Notebook or Jupyter Lab environment. Users can directly use and visualise the interactive plots of several ST platforms in the notebook.

Lastly, we provide the Code Ocean compute capsule to allow for reproduction of the results reported in this paper as well as for running the step by step analysis presented in the stLearn documentation.

### Reporting summary

Further information on research design is available in the Nature Portfolio Reporting Summary linked to this article.

## Data availability

Sequencing data, both raw and processed, for experimental datasets generated as part of this study, have been deposited to the NCBI's Gene Expression Omnibus (GEO) database under the accession code GSE236171 and made publicly available. All imaging data (e.g. imaging data from RNAscope and immunofluorescence-based histological studies) can be made available upon request. Sources and links to publicly available datasets, as described in the data section above, have also been provided. Briefly, deidentified patient gene expression data only, not the sequences, were downloaded from publicly available datasets (accessed February 2022). These include The Human Breast Cancer Visium dataset from the 10X Genomics website (https://support.10xgenomics.com/spatial-gene-expression/datasets/1.0.0/V1_Breast_Cancer_Block_A_Section_1; seqFISH+ data[13] from (https://github.com/CaiGroup/seqFISH-PLUS/blob/master/sourcedata.zip); Slide-seq data[15] for the mouse hippocampus from (https://www.dropbox.com/s/cs6pii5my4p3ke3/mouse_hippocampus_reference.rds?dl=0); 10x Genomics' public Mouse Brain Visium dataset from [https://support.10xgenomics.com/spatial-gene-expression/datasets/1.1.0/]; and Visium human brain dataset from Maynard et al.[11] [https://github.com/LieberInstitute/HumanPilot]. The data analysis complied with the terms and conditions of the data sources mentioned above. Our simulated dataset can be reproducibly generated using code available on the stLearn manuscript GitHub page (https://github.com/BiomedicalMachineLearning/stlearn_manuscript/tree/main/Main_figure_4_5_CCI_with_Sup/scripts/X6_breast_cancer_simulation). Source data are provided with this paper.

## Code availability

stLearn software is implemented using Python and the source code is available at https://github.com/BiomedicalMachineLearning/stLearn; detailed tutorials can be found at https://stlearn.readthedocs.io/. The public Github repository of the interactive version of stLearn, i-stLearn, can be accessed at https://github.com/BiomedicalMachineLearning/stlearn_interactive. Code to reproduce figures presented in this paper is available at https://github.com/BiomedicalMachineLearning/stlearn_manuscript.

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

## Acknowledgements

We thank all members in Nguyen's Genomics and Machine Learning Lab and Prof Matthew Ritchie at the WEHI Medical Research Institute for helpful discussions. This work has been supported by the Australian Research Council (ARC DECRA grant DE190100116 and DE150101578 to Q.H.N. and J.V., respectively), National Health & Medical Research Council (NHMRC Project Grant 1124503, 1163835, 2001514 to J.V., M.J.R. and Q.H.N. respectively), NHMRC Investigator Grant (GNT2008928 to Q.H.N.), a Wings for Life Project Grant and Fellowship (to M.J.R., Q.H.N. and L.F.G., respectively), a Senior Medical Research Fellowship from the Sylvia and Charles Viertel Foundation (2020001416 to JV), The University of Queensland (UQ), and the UQ Genome Innovation Hub. We further thank the staff of The University of Queensland Biological Resources Facility for the breeding and maintenance of animals used in this study. Imaging was performed at the Queensland Brain Institute Advanced Microscopy Facilities, using a Diskovery spinning disk confocal microscope and stereology microscope supported by the Australian Government (ARC LIEF grant LE100100074).

## Author contributions

Q.H.N., M.J.R., D.P., X.T., J.X. and B.B. conceived experiments and developed the algorithms. D.P., X.T., J.X., B.B. and Q.H.N. wrote the software. D.P., X.T., J.X., S.Y., T.M., B.B, Q.H.N., P.Y.L., L.F.G., E.F.W., J.V. and M.J.R. conducted experiments and analysed data. J.V. and M.J.R. annotated brain tissue, and A.R., S.L. and P.K.C. annotated cancer tissues. Q.H.N., D.P., J.X, B.B., L.F.G. and M.J.R. wrote the manuscript. All authors have reviewed and approved the manuscript.

## Competing interests

The authors declare no competing interests.
