## [Peer Review File · Nature Communications]

nature portfolio

Peer Review FileReviewer comments

Reviewer #1 (Remarks to the Author):

In this manuscript, Pham et al. present the stLearn method to 1. identify spatial trajectories of transcriptional states in ST data, 2. identify spatial cell-cell communication (CCC) and 3. combine transcriptomic and morphological information to impute expression in ST data. The method integrates gene expression, spatial distance and tissue morphology and shows better performance than traditional scRNA-seq methods without this integration. Applied on multiple ST datasets, the authors demonstrated the ability of stLearn and its superiority over traditional scRNA-seq methods on identifying meaningful spatial trajectories, CCCs and imputing missing ST data in development and disease progression contexts.

Overall, this manuscript addressed several important problems in the field of ST analysis. However, there are some major points remaining in the manuscript that should be addressed before the manuscript will be suitable for publication, most notably relating to improving benchmarking against not only non-spatial scRNA-seq methods but also alternative spatial approaches, as well as minor points that would also improve the clarity of the manuscript. Below are detailed comments:

Major concerns:

1. The computation of PSTS clearly depends significantly on the weighting parameter ω . While an ad hoc method for choosing a value of ω is presented, this should be expanded to more clearly show how the stLearn method balances spatial and gene expression information. In particular, is this optimal omega fairly similar across different datasets? Also, can it be quantified that the omega chosen through eqn. 10 outperforms other choices?
2. The use of morphological information and its contribution to the method should be more expanded upon in the results. In particular, the value of using a complicated ResNet model trained on different data over a simpler approach for capturing morphological data should be addressed in the manuscript, by adding comparisons to alternative versions of SME with a simpler form of the I matrix, and one without considering morphological information at all.
3. The comparison of SME clustering on the human cortex data should report values from all methods on all slices – in particular, SI Fig. 19e should include results from BayesSpace/SpaGCN, to contextualize its performance against other spatial methods.
4. The authors compare only to non-spatial pseudotime methods – it would be valuable to also perform a spatial comparison, such as to the SpaceFlow method (Ren et al., Nat. Com. 2022), to highlight the contributions of stLearn in particular.
5. Similarly, the CCC inference analysis only contains comparisons to other methods that do not incorporate spatial information. There exist prior methods in the literature that infer CCC using spatial information, such as SpaOTsc (Cang and Nie, Nat. Com. 2020) and SpaTalk(Shao et al., Nat. Com. 2022). Similarly to the pseudotime analysis, the comparison on spatial CCC inference should be extended to highlight how the manner in which stLearn considers spatial information compares to alternative approaches.

Minor concerns:

6. The authors should clarify the manner in which the framework is particularly “interpretable.”
7. In Fig. 1b and Fig. 1e, the notation for spatial relationship (S) are inconsistent with other panels and the descriptions in caption, which is D.
8. Results section should include information on how the clustering of TBI dataset shown in Fig. 2c is generated.
9. Fig. 2i – it is unclear what the various lines represent here.
10. Details of the leave-out simulation strategy for evaluating the SME method should be added to the methods section.
11. It is unclear what data and analysis is reflected in Fig. 6e.
12. In pseudo-code of Algorithm 1, a typo in line 6, should be $\text{Length}(mi) < \text{threshold}$.
13. Page 15: when writing “0 to 200 distance”, the appropriate unit of distance should be added.
14. In the section Spatial trajectory reconstruction, paragraph 3, there is a formula typesetting

error.

Reviewer #2 (Remarks to the Author):

Pham et al describe advances to their stLearn method that demonstrates use of spatial gene expression, puts weights on the distances between analyzed spatial measurements and integrated imaging (HE) information (or spatial information if a spot is in a neighborhood with similar gene expression values). They also developed a permutation test for L-R mapping using first order neighbors in the spatial data. The manuscript provides a good resource for the community and I would like to commend the authors for a very well written manuscript.

- 1) It would be great to see how the L-R results from stLearn compare to that of DIALOGUE (that uses a very different approach to map cell type specific cell-cell interactions) and GCGN (what uses graph to find higher order structures as compared to first order neighbors in stLearn)
- 2) stLearn currently does not involve data integration from multiple sections or conditions. This is a major bottleneck in current spatial data analysis. Can the authors demonstrate how common spatial pseudo-time trajectories could be inferred across sections (at the minimum) and how these patterns could be commonly annotated to increase interpretability across these sections? For example, if they see trajectory#1 in section#1 has similarities to trajectory#4 in section#5, how can one think of automatically annotating trajectory#1 and trajectory#3 as a "common" trajectory
- 3) Can the authors comment on why PSTS could be useful predicting trajectories (and by that biological patterns of expression) when one of the intermediate states is missing? For example, in Fig 3f, the authors only demonstrate some of the states which implies to the readers that cluster #16 jumps to cluster # 10 as the tumor progresses. However, this "jump" is across a very large actual distance. I believe this is just possibly needs a little bit more detail in the Results/Discussion section.
- 4) Can the authors demonstrate the accuracy of predicting missing values with stSME?

REVIEWER 1:

Overall, this manuscript addressed several important problems in the field of ST analysis. However, there are some major points remaining in the manuscript that should be addressed before the manuscript will be suitable for publication, most notably relating to improving benchmarking against not only non-spatial scRNA-seq methods but also alternative spatial approaches, as well as minor points that would also improve the clarity of the manuscript.

We thank the reviewer for this appraisal of our manuscript. Benchmarking against alternative spatial approaches has been included in the revised version of our paper. All other comments have also been addressed (see below).

Major concerns:

1. The computation of PSTS clearly depends significantly on the weighting parameter w . While an ad hoc method for choosing a value of w is presented, this should be expanded to more clearly show how the stLearn method balances spatial and gene expression information. In particular, is this optimal w fairly similar across different datasets? Also, can it be quantified that the w chosen through eqn. 10 outperforms other choices?

We thank the reviewer for these helpful comments, all of which have been addressed in the revised manuscript.

As requested, we have provided more clarification as to how the optimal w is chosen (see below and new subsection (red font) in Methods under '*Optimising of weighting parameter w in d_{PTS} calculations*'). This includes the description of a sensitivity analysis for the weighting parameter w where we controlled the relative contributions of spatial and gene expression information in three different spatial datasets (**Fig. S10a**); w values here ranged from 0 (spatial information only) to 1 (gene expression information only), with a step size of 0.01 between these extremes. In doing so, we show that finding the correct balance in the contribution of spatial distance and gene expression is indeed a quantitative process, and also that the optimal w is fairly similar (0.46-0.51) across these different datasets.

To further illustrate how an optimal w outperforms other choices (i.e., suboptimal weightings), we provide a case study for the breast cancer dataset where we compare and contrast the impact of different w values (that is, optimal vs. suboptimal weightings) on the spatial trajectories between a 'ductal carcinoma in situ' (DCIS) and invasive ductal carcinoma (IDC) states, as computed by PSTS (**Fig. S10b**). Whilst the gene expression data do contain the global transitional pattern of cells without the spatial connection (as observed in PCA/UMAP space), the gene expression distance (or global transitional pattern) is also correlated with the spatial distance between sub-clusters. Because of this, the global graph structures are similar, but quantitative changes (or differences) between them can be appreciated from the edge weights (**Fig. S10b, top row**).

Overall, the final graphs with $w=0$ (only spatial) and $w=0.46$ (optimal) have the same topology but the edges have different weights, favoring the latter (**Fig. S10b, bottom row**). Conversely, higher w values of

0.75 and 1 yield trajectories that have more branches and/or become a single root-like graph, which is not plausible biologically. Thus, the optimal $\omega=0.46$ best balances spatial and gene expression information, with the final and improved trajectory tree connecting the expected adjacency pairs for DCIS-IDC subclusters, and invasive IDC clusters (blue) via the shortest distance to a DCIS cluster (pink).

Lastly, it is important to highlight here that one of our original motivations for integrating spatial and gene expression information was to solve the possible issue of missing nodes (tissue spots), which could include intermediate nodes that are in the third dimension (depth) of the tissue. The relatedness of nodes present within the $\sim 2D$ spatial array is nonetheless captured by the gene expression data (and transitional patterns therein), which also helps solve issues around 2D perspective bias as shown in **Fig. S10c**.

In summary, the method for choosing the optimal ω is quantitative, and outperforms suboptimal weightings in that the final optimized PSTS trajectories are meaningful and do satisfy biological constraints, as demonstrated for both the spatiotemporal response of microglia to injury, and the established patterns of brain development.

The following text has been added to the revised manuscript to clarify and address the reviewer's comment here:

"The optimal ω is the one that balances the divergence from the two references, and its weighting value can be quantitatively determined through an ω sensitivity analysis where the contribution of spatial and gene expression components to corresponding PSTS trajectory graphs is assessed when ω changes from 0 to 1 (in 0.01 increments); this quantitative approach uses two reference graphs that use either only spatial information ($\omega=0$), or gene expression data ($\omega=1$) (see Fig. S10a). The point where the dissimilarity score is lowest optimally balances the individual contribution(s) of both the spatial and gene expression component towards the spatial trajectory being reconstructed. To also assess variation in the optimal ω parameter across different biological systems, we compared the determined optimal value for three independent ST-seq datasets (mouse TBI, mouse neurodevelopment, and human breast cancer progression), finding that the optimal ω was relatively similar (0.46-0.51; Fig. S10a). To also illustrate how the optimal ω outperforms other choices, we show that gene expression bias ($\omega>0.46$) can lead to non-specific trajectories, as exemplified in the DCIS-IDC case study (Fig. S10b) where a single DCIS (sub-cluster 6) now connects to either multiple or all IDC sub-clusters (Fig. S10b, right lower panels). This outcome goes against the findings of de Bruin et al. (2014) who reported evidence for spatially-branched evolution of cancer clones, with the multiple branches being derived from different clones (supported in our dataset by the transcriptional relationships/diversity within and between DCIS-IDC pairs). With spatial bias, on the other hand (e.g., when using spatial distance alone ($\omega=0$); Fig. S10b, left lower panel), we found a PSTS graph that appeared to accurately reflect the relationship(s) between DCIS and IDC clones (sub-clusters 6, 13, 16); this could be due to the edges from sub-clusters 6 and 16 in the initial graphs before optimisation already having had their weights (i.e., PSTS distance) stable, but these weights increased when ω changed from 0 to 0.46 (optimum). It is important to recognise here that spatial bias assumes nearby cells are more likely connected in the spatial progression process, which may lead to "over-fitting", that is, predicted branches only (and/or wrongly) connecting to the nearest cells; this "over-fitting" occurs because the spatial data of a thin $\sim 2D$ tissue section may contain "gaps" or "missing nodes" (Fig. S10c). As exemplified there, the tissue section may contain a gap between connected clusters involved in cell

movement/translocation, and where the intermediate nodes are not within the same section but rather the third dimension (depth) of the tissue block. In stLearn's PSTS, we utilize the gene expression information to capture this missing structure of the graph because the gene expression data still contains the global transitional pattern and/or relatedness of cells, even without the spatial connection (as observed in PCA/UMAP/Pseudotime latent space). Our optimization step for ω therefore overcomes issues associated with both sources of bias, with the identified optimal value of ω (0.46 in this example; Figure S10b) providing the right balance between the contribution of spatial and gene expression data, and a good fit to the actual spatial relationships that prevent trajectories from being misaligned with the biological process."

2. The use of morphological information and its contribution to the method should be more expanded upon in the results. In particular, the value of using a complicated ResNet model trained on different data over a simpler approach for capturing morphological data should be addressed in the manuscript, by adding comparisons to alternative versions of SME with a simpler form of the I matrix, and one without considering morphological information at all.

We thank the reviewer for these suggestions. As requested, we have expanded on demonstrating how morphological information positively contributes to outcomes with our stSME method. Further to what is now Fig. S19 in the revised manuscript, we also present new and/or additional data as part of **Fig. S20** and **Fig. S21**, demonstrating the benefits of using a convolutional neural network model, i.e. ResNet50, over a more traditional, simpler approach (handcraft features), again as requested.

In brief, we used H&E images from mouse brain ST dataset (and their corresponding annotations) to extract morphological information, either via ResNet50 (2048 features) or from handcraft image features (12 in total; **Fig. S20a,b**). For the latter, feature groups such as pixel color values (R, G, B) and those associated with cell nuclei (e.g. intensity, eccentricity/elongation, and area) were all positively correlated within (**Fig. S20b**). Importantly, handcraft features were not able to define specific regions as specific and/or accurate as the ResNet50 model, as shown in **Fig. S20c** in relation to e.g. white matter tracts, and also in **Fig. S20d** for injured brain tissue where only ResNet50 but not handcraft features could distinguish and separate lesioned cortex from spared (healthy) regions as well as the hippocampus on a reduced 3D tSNE space.

We also provide other additional evidence to show that the extra biological information from histological images, as extracted via ResNet50, is crucial for improving the performance of the stSME algorithm (**Fig S21a**). In brief, we compared and contrasted three scenarios where morphological information was not considered at all (baseline resulting from default Louvain clustering method; *scenario 1*), or where morphological information was considered, using either a simplified matrix for the stSME algorithm with the same handcraft image features shown in Fig S20a,b (*scenario 2*), or stSME with ResNet50 features (*scenario 3*). Note that the clustering results are noisier when handcraft features are used (**Fig. S21a, middle**) compared to both the original clustering (**Fig. S21a, left**) and stSME with ResNet50 features (**Fig. S21a, right**). It is also important to note that only stSME with ResNet50 features was able to accurately identify and separate smaller anatomical structures and/or sub-regions such as the dentate gyrus, the CA1

and CA3 regions. Use of stSME with ResNet50 features also yielded more accurate clustering for the breast cancer dataset (**Fig S21c, right**) compared to when handcraft features were used (**Fig S21c, middle**).

Lastly, and beyond clustering, we provide newly added evidence in **Fig S21b** that the use of ResNet50 features also yields superior performance in relation to imputing missing values. Specifically, stSME with ResNet50 features successfully rescued most missing values (relevant spots with “0” expression) for the marker genes *Pla2g2f* and *Lhfp1* that label the dentate gyrus and CA3 region, respectively (**Fig S21b, right**); imputation using handcraft image features rescued much fewer missing values (**Fig S21b, middle**).

Overall, these findings highlight the value of the ResNet model over simpler approaches, or those that do not incorporate morphological data at all, in relation to stSME’s clustering performance (less noise and improved ability to resolve anatomical structures), and also its ability to impute / recover missing values.

3. The comparison of SME clustering on the human cortex data should report values from all methods on all slices – in particular, SI Fig. 19e should include results from BayesSpace/SpaGCN, to contextualize its performance against other spatial methods.

Amended. The original Fig. S19e (**Fig S22e** in the revised manuscript) has been updated, with the reported values now being derived from all of the 12 slices available within this dataset, and also with inclusion of an expanded range of other methods, including BayesSpace and SpaGCN, as requested.

While clustering is not the main analysis goal and/or advance offered by stLearn’s SME module, the revised data clearly show comparable results with other state-of-the-art methods like BayesSpace and SpaGCN. That said, we do find that stSME results between different sections/tissues appear more consistent, which is visually reflected in the clustering plots for each of the 12 samples (added as **Fig. S23** to the revised manuscript). Indeed, a statistical test of the variance in the clustering performance of SMEclust was significantly smaller compared to both BayesSpace and SpaGCN (two-sided F-test <0.05), suggesting SMEclust performance to be more stable.

4. The authors compare only to non-spatial pseudotime methods – it would be valuable to also perform a spatial comparison, such as to the SpaceFlow method (Ren et al., Nat. Com. 2022), to highlight the contributions of stLearn in particular.

We thank the reviewer for this suggestion. The revised manuscript now includes a direct comparison to SpaceFlow, as suggested, and the results have been added to the revised manuscript (**Fig 2j** and **Fig S6**).

In brief, we find that SpaceFlow’s pseudo-Spatiotemporal Map (pSM) was able to generate spatially smooth gradients. However, the features of its deep graph networks led to the issue of multiple terminators (endpoints of the trajectory; **Fig 2j** and **Fig S6**). This may occur because of the spatial regularisation and/or loss of information in the latent space that is used by SpaceFlow to compute pseudotime scores. In our analysis, SpaceFlow’s pseudotime values did not form a pattern that then enabled the drawing of a tree for optimising a given trajectory (from low to high pseudotime scores), as done by stLearn. Because of this, and as reported in the original paper, SpaceFlow does not predict a trajectory, giving only gradient maps instead (pSM).

We further show that, without a clear difference in pseudotime values, transition genes and significant pathways associated with meaningful biological changes taking place across a given tissue are also less likely to be detected by SpaceFlow, as illustrated in **Fig S6b** for microglia activation following brain injury (the existence and pattern of which we externally validated through histological studies).

The relevant Results section of the revised manuscript now reads:

*“To also benchmark against other pseudotime methods that do use spatial information, we next compared the performance of PSTS to that of SpaceFlow (Ren et al., 2022)(**Fig. 2j** and **Fig S6**). While SpaceFlow’s pseudo-Spatiotemporal Map (pSM) did provide spatially smooth gradients (indicating a good variogram), the features of its deep learning framework led to multiple terminators (endpoints in the trajectory); SpaceFlow’s spatial regularisation and/or loss of information in the latent space after dimensionality reduction (which is used by SpaceFlow to calculate pseudotime scores) may have contributed to this issue. In our analysis, the pseudotime values computed by SpaceFlow did not form a pattern that enabled the drawing of a tree for optimising the trajectory from low to high pseudotime scores, as done in stLearn. The SpaceFlow result also did not allow us to identify transition genes along the trajectory, and the significant pathways associated with these (**Fig S6b**). Overall, PSTS thus outperformed all other trajectory inference methods tested, including the method that uses spatial information (**Fig 2j**).*

5. Similarly, the CCC inference analysis only contains comparisons to other methods that do not incorporate spatial information. There exist prior methods in the literature that infer CCC using spatial information, such as SpaOTsc (Cang and Nie, Nat. Com. 2020) and SpaTalk(Shao et al., Nat. Com. 2022). Similarly to the pseudotime analysis, the comparison on spatial CCC inference should be extended to highlight how the manner in which stLearn considers spatial information compares to alternative approaches.

We thank the reviewer for their suggestion to perform additional comparisons against other cell-to-cell communication / interaction (CCC/CCI) methods, specifically those that also do use spatial information. This has now been addressed in the revised manuscript. Specifically, we have performed the requested comparisons against both SpaTalk (Shao et al., Nat. Com. 2022), SpaOTsc (Cang and Nie, Nat. Com. 2020), as well as NCEM, which also uses a graph-based method (Fischer et al., Nat. Biotech.,2022), and the results have been added to **Fig 5**. Our original conclusions remain unchanged in that we still find that stLearn’s SCTP interaction analysis outperforms all other methods with regards to reducing false positive interactions between ligand-receptor pairs and cell types. This applied to both the simulated dataset (**Fig 5c,d**) and the breast cancer dataset where interactions that should not have occurred and/or are biologically implausible based on the physical distance were still predicted (**Fig 5e,f**).

In summary, stLearn’s SCTP method, which can be applied to various datasets, is robust in detecting communication between cells, and it advances from other methods by reducing false positive interactions through the integration of spatial information and statistical tests.

Minor concerns:

6. The authors should clarify the manner in which the framework is particularly “interpretable.”

Amended and apologies for not making this clearer. The integration framework is interpretable as the contribution of tissue morphology, gene expression and spatial distance can be quantified. Particularly, we compute the contribution of spatial distance and gene expression to the PSTS score in trajectory analysis (equation 3), the spatial neighbourhood and ligand-receptor expression in the SCTP method for cell-cell interaction analysis (equation 13), and for the integration of the three datatypes in stSME for imputation (equation 25). We have added the following sentence to the revised manuscript to clarify what we meant by the framework being “interpretable” (see below and Results, paragraph 1, red text):

“...; this graph-based framework is interpretable as the individual contribution of each type of information can be quantified.”

7. In Fig. 1b and Fig. 1e, the notation for spatial relationship (S) are inconsistent with other panels and the descriptions in caption, which is D.

Apologies for this oversight and any inconvenience / confusion caused by this inconsistency. Spatial distance is now consistently referred to as “D”, including for Fig. 1b and 1e.

8. Results section should include information on how the clustering of TBI dataset shown in Fig. 2c is generated.

Amended. The following information has been added to the Results section of the revised manuscript:

“We then applied SME-based clustering (see Methods) to segment the brain (Fig. S2 and Fig. 2c), using the Allen Mouse Brain Atlas for the fine-tuning of clustering parameters (Lein et al., 2007),...”

In addition, we also added the following subsection to the Methods (highlighted again in red font):

“Clustering of TBI dataset

PCA and standard Louvain clustering using the top 50 PCs were used to detect 15 broad clusters across the TBI sample (Fig. S2 and Fig. 2c). Clusters were split further if they were spatially separated within the tissue using the stLearn sub-clustering option. Each cluster was annotated based on comparisons with well-defined anatomical regions given by the Allen Mouse Brain Atlas reference (Lein et al., 2007). Data were visualised in both UMAP and ForceAtlas2 space (Jacomy et al., 2014)(Fig. S2).”

9. Fig. 2i – it is unclear what the various lines represent here.

Amended. We have revised and added this information to the relevant section of the legend for Figure 2:

*“Variogram plots depicting the autocorrelation of PSTS/pseudotime values for each spot. Once each pair of locations is plotted, a model (**green line**) is fitted through them. These plots show the spatial variance of PSTS/pseudotime values produced by stLearn PSTS, Slingshot and Monocle 3. Lower values of the semi-variance Matheron estimator suggest a higher PSTS/pseudotime continuity in the spatial context, and thus a more likely trajectory (refer to Methods); **PSTS semi-variance is indicated by the red dashed line.**”*

10. Details of the leave-out simulation strategy for evaluating the SME method should be added to the methods section.

Amended. We added the text describing this strategy to the Methods:

“Leave-out simulation for evaluating SME imputation.

Two simulation approaches were developed to evaluate the performance of the SME imputation method; both used the raw gene expression matrix as the ground truth.

In the first approach, 20% of all data points were randomly selected and replaced with a "0" to simulate missing values. stSME imputation was then applied to recover these missing values, followed by Louvain clustering. Performance of the stSME imputation algorithm was evaluated by using the adjusted Rand index (ARI), assessing clustering results against the ground truth (i.e. original clustering obtained from the full data).

In the second approach we iterated through every spot on the tissue slide, treating each as being without gene expression values to simulate missing spot scenarios. The SME method was then again applied to impute the top 2000 highly variable genes (HVGs). After all spots were imputed, the spatial autocorrelation of the imputed gene expression matrix was calculated against the original gene expression matrix for each HVG to evaluate the performance of the stSME imputation.”

11. It is unclear what data and analysis is reflected in Fig. 6e.

Apologies for not making this clearer. Results presented in **Fig. 6e** relate to the outcome of the simulation approach shown in **Fig. 6d**. We have amended the figure legend, the relevant section of which now reads:

“e, Box plot showing poorer clustering results when the SME option is not used, as assessed by adjusted Rand index (ARI); results are derived from the simulation approach shown in d, which was repeated 10 times to assess robustness of the stSME imputation approach, and using the original clustering with full data as the reference.”

12. In pseudo-code of Algorithm 1, a typo in line 6, should be $\text{Length}(mi) < \text{threshold}$.

Apologies for the oversight and thank you for spotting this. We have corrected this in the revised manuscript.

13. Page 15: when writing “0 to 200 distance”, the appropriate unit of distance should be added.

Thank you again for spotting this. The unit of distance has been added in the revised manuscript.

14. In the section Spatial trajectory reconstruction, paragraph 3, there is a formula typesetting error.

Apologies for this oversight. This has been fixed in the revised manuscript.

REVIEWER #2:

Pham et al describe advances to their stLearn method that demonstrates use of spatial gene expression, puts weights on the distances between analyzed spatial measurements and integrated imaging (HE) information (or spatial information if a spot is in a neighborhood with similar gene expression values). They also developed a permutation test for L-R mapping using first order neighbors in the spatial data. The manuscript provides a good resource for the community and I would like to commend the authors for a very well written manuscript.

We thank the reviewer for their positive appraisal, and also the time they have taken to review our manuscript and for offering suggestions to further improve our manuscript.

1. It would be great to see how the L-R results from stLearn compare to that of DIALOGUE (that uses a very different approach to map cell type specific cell-cell interactions) and GCNG (what uses graph to find higher order structures as compared to first order neighbors in stLearn)

We thank the reviewer for these suggestions, which reflect a similar request from Reviewer 1 (see point 4) in relation to performing some additional benchmarking. We have addressed this in the revised manuscript through further comparisons as to how stLearn's CCI module performs against other spatially-based cell-cell communication analysis methods.

We tried to run GCNG (<https://github.com/xiaoyeye/GCNG>; Yuan and Bar-Joseph, 2019), but the source code had errors (also reported by other users) and this issue has not been fixed for 2 years. We then modified the code so that we could at least run and compare the GCNG method against stLearn. The log of the training run for the human breast cancer ST dataset, with very low performance, is included in Appendix 1 at the end of this letter. Importantly, because GCNG did not work due to technical issues in the source code, we sought and included another graph-based method, namely NCEM, to further accommodate the reviewer's request here. The results for the NCEM analysis have been added to **Fig. 5d, f** in the main manuscript, along with SpaTalk and SpaOTsc (the additional methods requested by reviewer 1). Our original conclusions remain unchanged here in that stLearn's SCTP interaction analysis outperformed all other methods with regards to reducing false positive interactions between ligand-receptor pairs and cell types.

In relation to DIALOGUE (<https://github.com/livnatje/DIALOGUE>), this method performs a dimensionality reduction (sparse latent covariates) to identify multicellular programs (MCPs) of different coordinated gene expression programs, from different cell types and their regulated genes. This method does not directly test for significant LR interactions and/or pairs, rather the resulting MCPs only include genes that were either up- or downregulated in a given cell-type within the MCPs; these genes may or may not be LR pairs. Indeed, when we completed a DIALOGUE run for two merged groups, namely cell-type-1 as basal (basal_like_1 and basal_like_2) and cell-type-2 as immune (stroma, B cell, T cell, and macrophages), the analysis resulted in only one MCP with very few genes in each cell-type showing as differentially expressed, and these genes did not appear as LR pairs. DIALOGUE was also not designed to identify MCPs for one versus another cell type. A direct comparison of stLearn's SCTP method against DIALOGUE is therefore not possible, but we have highlighted the above-outlined differences in approach and/or their purpose in the Discussion of the revised manuscript (see red text in CCI section), which reads:

“stLearn also differs from other methods such as DIALOGUE \cite{} that focus on identifying coordinated gene expression programs in different cells based on regulated genes, but that do not directly test for significant interacting LR pairs; the multicellular programs used by DIALOGUE only include genes that are up- or downregulated in different cell types, and these may or may not be LR pairs”

2. stLearn currently does not involve data integration from multiple sections or conditions. This is a major bottleneck in current spatial data analysis. Can the authors demonstrate how common spatial pseudo-time trajectories could be inferred across sections (at the minimum) and how these patterns could be commonly annotated to increase interpretability across these sections? For example, if they see trajectory#1 in section#1 has similarities to trajectory#4 in section#5, how can one think of automatically annotating trajectory#1 and trajectory#3 as a “common” trajectory?

We thank the reviewer for this excellent question / suggestion, which we agree is an important one and a major bottleneck. The revised manuscript proposes a data integration approach for spatial trajectory analysis where users can automatically identify shared driver genes that are statistically associated with pseudotime scores of trajectories that represent the same biological process across different datasets. This integration of multiple datasets is useful for identifying trajectories that have consistent patterns and are thus stable. By using shared driver genes, the integration approach does not necessarily require tissue sections to be registered and therefore can be broadly applied.

In instances where sections are adjacent or mappable to one to another, users can project those sections to a common coordinate framework (e.g., registration methods like geometric transformation to map moving sample to a reference - see **Fig. S13a,b**). Post-registration, the location of spots and/or cells belonging to the same cluster should be next to each other in the common coordinate system. PSTS can be directly applied to this merged dataset, as shown in **Fig. S13c** of the revised manuscript.

In cases where sections cannot be transformed to a common coordinate framework, like tissue sections collected from different samples for different conditions, or between sections that have gaps and/or intermediate states, we offer the following strategy to solve this issue. For each section, the spatial trajectory is to be applied first. Next, the transition gene markers for each trajectory can be compared and the intersection proportion between two or multiple sections calculated, that is, the consensus set of markers identified from a correlation analysis between gene expression and pseudotime scores along the trajectory. This set of gene markers is then used for identifying the common (or shared) trajectories.

To demonstrate feasibility and usefulness of the latter approach, we applied this also to two adjacent sections available from 10X Genomic’s breast cancer block A (hereafter referred to as sections 1 and 2). The results have been included as **Fig. S13d-h** of the revised manuscript. In brief, applying our PSTS method to sections 1 and 2 yielded similar patterns of spatial trajectories that reflect the progression of ‘ductal carcinoma in situ’ (DCIS) to invasive ductal carcinoma (IDC) states in three different clades (**Fig. S13d**). The trend of pseudo-time-space values for the DCIS, that is, middle to edge, and then on to the connected IDC, was also consistent between sections (**Fig. S13e**). A correlation analysis between gene expression and pseudotime scores along the trajectory was then used to identify overlapping transition markers, which revealed a consensus set between the two sections (**Fig. S13f**, see Venn diagrams).

As can be appreciated from the histograms in **Fig. S13f**, correlation scores for the genes that were not consistently detected between the samples mostly followed the expected pattern of up- (positive correlation) and downregulation (negative correlation), but they did not pass the correlation threshold of 0.4 or -0.4, respectively (blue and yellow histograms). Red histograms consistently show stronger correlation scores for the shared driver genes. We then performed downstream enrichment analysis for up- and downregulated gene transition markers, that either were or were not intersected between sections 1 and 2 (**Fig. S13g,h**). We observed that regulated pathways relating to extracellular matrix organisation (and thus tumor progression) were highly conserved across the sections when analysing upregulated transition marker genes (**Fig. S13g**). For downregulated genes, we consistently found that pathways relating to apoptosis were significant in every case (top-5 for both intersected (shared) and non-intersected genes in section 2, 7th GO term for section 1; **Fig. S13h** and data not shown); the combination scores and significance values were lower here (compared to upregulated pathways) due to the genes involved being only a minority of the pathway gene components. Overall, we observed a consensus in the spatial pattern and also a shared proportion of transitional markers across the trajectories that are likely key in terms of biological significance. The relevant added text in the Results section of the revised manuscript (marked up in red font) reads:

“Lastly, to also examine here if spatial trajectories translate and/or can be inferred across different tissues, we devised a broadly applicable integration strategy. This integration strategy harnesses the power of multiple spatial datasets for identifying trajectories that have consistent patterns and are thus stable. Users can either register two (or more) tissues into a common coordinate framework and then run PSTS on the merged dataset (Fig. S13a-c). Where registration is not possible, users can run PSTS independently and identify shared driver genes that are consistently associated with common trajectories between the tissues (Fig. S13d-h); as the latter approach does not require sections to be registered, it can be broadly applied. To demonstrate applicability of both approaches, we took advantage of ST data here from an adjacent tissue section of the same breast cancer sample. We demonstrate that registering sections to a common coordinate framework identifies and/or confirms the common trajectories between sections (Fig. S13a-c). We further show that, even without image registration, the identification of shared driver genes and subsequent enrichment pathway analysis can support and annotate the common trajectory between sections (Fig. S13f-h). Thus, PSTS can effectively infer spatial trajectories across different sections through visualization pattern matching, and by making use of shared transition markers.”

We indeed observed very similar patterns of spatial pseudo-time trajectories from DCIS to IDC states for the three different clades in each section (Fig. S13d); the trend of pseudo-time-space values within a clade was also consistent (Fig. S13e).

A related statement highlighting has also been added to the PSTS section (in red font) of the Discussion of the revised manuscript:

“...Lastly, for instances where integration of data from multiple sections and/or conditions is required (a major bottleneck in the field that is beginning to be addressed; Long et al., 2023) for spatial trajectory analysis, we offer the following recommendations and/or workarounds. For adjacent sections (replicates from the same tissue block), these can be transformed into a common coordinate framework to form either a large 2D spatial array by expanding the original matrix, or a 3D spatial matrix by adding layers into the

original data. This merged dataset can then be used directly for stLearn's PSTS trajectory analysis (Fig. S13c). For sections that cannot be transformed into a common coordinate framework, such as those representing samples from different biological and/or treatment conditions, we suggest applying the spatial trajectory method to each section first and then finding the shared transition markers between trajectories (Fig. S13f). By calculating the intersection proportion between two (or more) sections based on these markers, it is then possible again to identify and/or annotate the common trajectories as well as to independently confirm their existence."

Taken together, the revised manuscript now provides both evidence and examples as to how stLearn's PSTS method can be utilised to effectively address the issue of inferring trajectories across different sections, that is, through visualization pattern matching and by identifying transition markers.

3. Can the authors comment on why PSTS could be useful predicting trajectories (and by that biological patterns of expression) when one of the intermediate states is missing? For example, in Fig 3f, the authors only demonstrate some of the states which implies to the readers that cluster #16 jumps to cluster # 10 as the tumor progresses. However, this "jump" is across a very large actual distance. I believe this is just possibly needs a little bit more detail in the Results/Discussion section.

We thank the reviewer for this question and have adopted their suggestion to add some more detail to the Discussion to clarify where and/or how PSTS is still useful for predicting trajectories when an intermediate state is missing. As also highlighted in our response to Reviewer 1 (point 1), the issue of missing intermediate states and/or regions was actually what inspired us to combine both spatial and gene expression. Specifically, in the example of breast cancer progression that the reviewer refers to here (Fig. 3f), it is well established that the invasive cancer can move to form cancer clones outside the ducts. As the "connection" for this move may appear and/or be present elsewhere in the larger 3D space of the tumor biopsy sample, such a move can appear as a gap (or jump) within the confines of a thin ~2D tissue section (Fig. S10c). Experimentally, only a 3D spatial technology could fully solve this problem. However, in absence of that, PSTS can help solve this conundrum by using the gene expression relationships to connect spots that are not spatially next to each other.

We have added the sentences to the Discussion of the revised manuscript to clarify:

"By balancing spatial with gene expression information, our PSTS method can also help solve trajectories when one of the intermediate states may be missing. Specifically, where an intermediate node or state is missing from the section (i.e., when there is no direct and/or obvious spatial connection), the relatedness of other spots / nodes in the trajectory can still be inferred from the global transitional pattern within the gene expression information (as observed in PCA/UMAP/Spseudotime latent space)"

In addition, we also further mention and/or clarify this issue in the Methods section of the revised manuscript where we now more extensively discuss the process for optimising the value of the weighting parameter ω by adding the following text:

"It is important to recognise here that spatial bias assumes nearby cells are more likely connected in the spatial progression process, which may lead to "over-fitting", that is, predicted branches wrongly connecting to the nearest cells; "gaps" or "missing nodes" within the spatial data of a thin ~2D tissue

section may pose an additional issue for inferring continuous trajectories. Indeed, as exemplified in Fig. S10c, a tissue section may contain a gap between connected clusters involved in cell movement/translocation in instances where the intermediate node is not within the same section but rather the third dimension (depth) of the tissue block, causing perspective bias."

It is important to recognise here that spatial bias assumes nearby cells are more likely connected in the spatial progression process, which may lead to "over-fitting", that is, predicted branches wrongly connecting to the nearest cells; "gaps" or "missing nodes" within the spatial data of a thin ~2D tissue section may pose an additional issue for inferring continuous trajectories. Indeed, as exemplified in Fig. S10c,

We trust that this alleviates the expressed concern.

4. Can the authors demonstrate the accuracy of predicting missing values with stSME?

This is shown in **Fig. 6e** of the revised manuscript where we used a simulated mouse brain ST dataset to compare clustering performance with and without stSME imputation. Since there is no definitive way to determine if a gene or spot has missing values, we treated the raw gene expression matrix of the Visium mouse brain ST dataset as the ground truth. We then randomly selected 20% of the values in the raw expression matrix and replaced them with a "0" to represent missing values. Next, we applied stSME to recover these missing values, followed by Louvain clustering. Overall, the clustering results with stSME imputation showed much less noise and the recovery of missing values again permitted for sub-regions of the hippocampus (i.e., CA1 from CA3) to be resolved (**Fig. 6d**). To test for accuracy, we repeated the simulation-imputation process 10 times and used the original clustering as the reference to calculate the ARI score. The results show that stSME imputation consistently resulted in a higher ARI compared to no imputation, suggesting that the imputation is both accurate and reliable.

We corroborated these findings by also testing stSME's imputation method on the Visium human breast cancer dataset (block A, sections 1 and 2) in order to also evaluate its performance between samples and/or biological replicates. Here, using the two samples and/or replicates (i.e., sections 1 and 2), we iterated through every spot on each tissue slide, treating each spot as 'without the gene expression' (leaving out all original gene expression values), and then applied the stSME method to impute the top 2000 highly variable genes (HVGs) based on expression data from the surrounding spots. After all spots were imputed, we compared the spatial autocorrelation (Moran's I index) of the imputed gene expression matrix against the original gene expression matrix for each HVG. We then visualized the results in a scatter plot for all spots and all genes (**Fig. 6f** in the revised manuscript), with the x-axis representing the spatial autocorrelation for replicate 1, and the y-axis representing the spatial autocorrelation in replicate 2 for the same genes; data points are the spatial autocorrelation for each HVG based on the imputed value from n iterations (with n being the number of spots). The colour-coding reflects the overall percentage of missing values for that particular HVG (i.e., sparsity of the gene) in the original UMI count matrix. The overall trend shows that the stSME imputation method performs similarly robust across the two replicates, and also that there is better imputation performance for HVGs that are less sparse (lower

percentage of 0 gene counts), which is logical and expected because of the increased richness of information in the surrounding spots.

APPENDIX 1

Log of the GCNG training run (with modified code) for the Visium human breast cancer ST dataset (10x Genomics).

```
Epoch 96/100
75/75 [=====] - ETA: 0s - loss: 0.6889 - acc: 0.5779
Epoch 96: val_acc did not improve from 0.56355
75/75 [=====] - 15s 205ms/step - loss: 0.6889 - acc: 0.5779 - val_loss: 0.6924 - val_acc: 0.5619
Epoch 97/100
75/75 [=====] - ETA: 0s - loss: 0.6888 - acc: 0.5777
Epoch 97: val_acc did not improve from 0.56355
75/75 [=====] - 12s 164ms/step - loss: 0.6888 - acc: 0.5777 - val_loss: 0.6924 - val_acc: 0.5619
Epoch 98/100
75/75 [=====] - ETA: 0s - loss: 0.6888 - acc: 0.5769
Epoch 98: val_acc did not improve from 0.56355
75/75 [=====] - 14s 184ms/step - loss: 0.6888 - acc: 0.5769 - val_loss: 0.6924 - val_acc: 0.5619
Epoch 99/100
75/75 [=====] - ETA: 0s - loss: 0.6888 - acc: 0.5774
Epoch 99: val_acc did not improve from 0.56355
75/75 [=====] - 13s 179ms/step - loss: 0.6888 - acc: 0.5774 - val_loss: 0.6924 - val_acc: 0.5619
Epoch 100/100
75/75 [=====] - ETA: 0s - loss: 0.6888 - acc: 0.5778
Epoch 100: val_acc did not improve from 0.56355
75/75 [=====] - 14s 189ms/step - loss: 0.6888 - acc: 0.5778 - val_loss: 0.6924 - val_acc: 0.5619
```

Reviewer comments, second round

Reviewer #1 (Remarks to the Author):

In this revision the authors have added new results demonstrating benchmarking of the stLearn method against other ST analysis methods, as well as more detailed investigation of the contribution of the pre-trained ResNet50-based model. In doing so, they have addressed most of the previously raised concerns about the manuscript. However, there remain some concerns primarily regarding the methodology used in benchmarking stLearn on the task of spatial pseudotime analysis, relating to major concern #4 raised previously. The essence of the benchmarking made for stLearn PSTS is that the one other spatial pseudotime method, SpaceFlow, doesn't work on the dataset. However, the current qualitative comparison fails to answer the question of whether the difference is actually due to difference in the capability of the methods, or just that the settings are less amenable to SpaceFlow than to PSTS, and so it does not succeed as a meaningful benchmark. Upon addressing these points, the manuscript will be suitable for publication.

Major concerns:

1. In every case where stLearn is compared to an alternative method, the methods section should include a description of how the alternative method was used, any parameter settings that were chosen, etc.
2. It appears that a pre-selected route of 4->2->3->1 is essentially prescribed in the PSTS analysis. What is the point of the PSTS/etc. if you choose the trajectory by hand, and how dependent are the results on this choice?
3. Why can't the spatial PAGA graph just be computed over all of the clusters so that whatever route fits best can be determined from the data?
4. Similarly, wouldn't it make more sense to test other methods using their own clusters instead of the stLearn ones?

Minor concerns:

5. There is some funny typesetting in the methods presumably due to underscores in num_exp_genes being typeset incorrectly.
6. There are several broken references (Fig. ??, etc.) in the manuscript.

Reviewer #2 (Remarks to the Author):

The reviewers believe the authors addressed their concerns to the best of their abilities and given the timeframe and recommends this manuscript for publication.

REVIEWER COMMENTS

REVIEWER 1 (Remarks to the Author):

In this revision the authors have added new results demonstrating benchmarking of the stLearn method against other ST analysis methods, as well as more detailed investigation of the contribution of the pre-trained ResNet50-based model. In doing so, they have addressed most of the previously raised concerns about the manuscript. However, there remain some concerns primarily regarding the methodology used in benchmarking stLearn on the task of spatial pseudotime analysis, relating to major concern #4 raised previously. The essence of the benchmarking made for stLearn PSTS is that the one other spatial pseudotime method, SpaceFlow, doesn't work on the dataset. However, the current qualitative comparison fails to answer the question of whether the difference is actually due to difference in the capability of the methods, or just that the settings are less amenable to SpaceFlow than to PSTS, and so it does not succeed as a meaningful benchmark. Upon addressing these points, the manuscript will be suitable for publication.

We thank the reviewer for this positive appraisal, and also the additional comments to further improve the clarity and impact of our study. Our revisions now fully address the issues that were raised around the benchmarking of PSTS against SpaceFlow. Below, we provide a point-by-point response to each of the comments raised and trust that these now satisfy the reviewer.

MAJOR CONCERNS

1. In every case where stLearn is compared to an alternative method, the methods section should include a description of how the alternative method was used, any parameter settings that were chosen, etc.

We highly value code and data reproducibility, and our paper contains reference to two well documented github sites specifically for this purpose, that is, one for broad usage of the software, and another for reproducing *all* figures in the manuscript:

1. https://github.com/BiomedicalMachineLearning/stlearn_interactive
2. https://github.com/BiomedicalMachineLearning/stlearn_manuscript

We have also included a detailed description as to how alternative methods were used in comparisons to stLearn and specified parameter settings, as requested. The relevant section of the Results now contains a reference to both the Methods and Supplementary Note 1 (see section 1.2, highlighted in red font for your convenience), to refer the reader to this detail.

Lastly, whilst comparing between methods, we noted (as shown in **Fig. 2**, and described below) the unique ability of PSTS to reconstruct trajectories within the spatial context, something that other methods are not designed to do. We describe this feature further under points 2 and 3 below.

2. It appears that a pre-selected route of 4->2->3->1 is essentially prescribed in the PSTS analysis. What is the point of the PSTS/etc. if you choose the trajectory by hand, and how dependent are the results on this choice?

We apologise for any confusion caused here and now better highlight the unsupervised nature of PSTS in the revised manuscript. Specifically, our PSTS trajectory prediction is an automated process without human input, and we can confirm therefore that the route of 4->2->3->1 was *not* pre-selected. Rather,

PSTS determined this route to be the shortest path from an unbiased set of all possible paths that move from the node with the lowest PSTS score (i.e., 4) to the one with the highest PSTS score (i.e., 1); nodes 4 and 1 are automatically defined on that basis. The unsupervised process for identifying the optimal path between nodes 4 and 1 involved four sequential, automated steps:

- Construction of a spatial PAGA graph based on the gene expression data and all cluster labels (**step 1**). This is a fully connected graph, where every node is initially connected to all other nodes. Nodes in the PAGA graph are unsupervised stLearn clusters, with the edges representing distance measures calculated from both spatial coordinates and gene expression data (Algorithm 1 - Main manuscript).
- Create a set of all possible paths from cluster 4 to cluster 1 (that is, from the lowest to the highest PSTS score) based on the spatial-PAGA graph (**step 2**). In this instance, this yielded a total of 170 possible paths to travel from cluster 4 to cluster 1 in the initial graph. The top-15 shortest paths connecting node 4 to node 1 are presented below; these results have also been included in the revised Fig. S6a,b.
- Calculate and rank each of the possible paths based on the physical spatial distance (**step 3**). Note that the shortest path resulting from the above steps was found to be 4->2->3->1, with a total distance of ~1578 pixels; we validated this path in the current paper. To our understanding, the PSTS shortest path finder is the first to utilise both the gene expression graph and spatial distance map to search for, and in an unsupervised way, the most likely trajectory connecting two states/locations in a tissue section.
- Finally, for visualisation, the optimal path with the shortest spatial distance is selected, while the other connections (graph edges) are subsequently trimmed (**step 4**).

A description of the above process has also been added to the main text and Supplementary Note 1 (Section 1.3).

The code implementation is available at:

https://github.com/BiomedicalMachineLearning/stLearn/blob/master/stlearn/spatials/trajectory/shortest_path_spatial_PAGA.py

Figure: The top-15 paths based on the shortest physical spatial distance from cluster 4 to cluster 1, as determined by the minimum spanning tree algorithm. Paths are ranked based on the total spatial distance between the nodes. The highlighted result of 1577.59 px represents the shortest distance between clusters 4 and 1, and was hence considered the optimal (validated) path.

3. Why can't the spatial PAGA graph just be computed over all of the clusters so that whatever route fits best can be determined from the data?

As clarified under point 2 above, this is precisely what happens. The PSTS method automatically determines the best fit route from all possible paths connecting all the clusters. The best fit route (i.e. 4->2->3->1) was entirely determined from the data, without (pre-)selecting a sub-graph or path from the PAGA graph.

We have otherwise also extended the downstream analysis of path-defining (transition) genes, using the global spatial PAGA graph results described above. We again find that the optimised route, as identified by PSTS, produced biologically meaningful results in terms of the number and relevance of transition genes (n=121) and pathways detected (those significantly associated with the scores across the whole tissue section), and also the relationship of these to microglia activation (a process that we experimentally validated). They included microglia activation markers C1qa, C1qb, Tyrobp, and Fcgr1g, and also relevant top-ranked pathways "microglia pathogen phagocytosis pathway", "TYROBP causal network", and "oxidative damage/stress" (Fig. S6c). As per previous, PSTS outperformed SpaceFlow here in the number of transition genes detected and, in sharp contrast to PSTS, SpaceFlow did not detect any microglia-related pathways in the enrichment analysis. One possible reason for this discrepancy may relate to over-smoothing by SpaceFlow, as already commented on in the previous revision.

4. Similarly, wouldn't it make more sense to test other methods using their own clusters instead of the stLearn ones?

To clarify, we had already implemented SpaceFlow's own clustering result in the original analysis as part of the `pseudo_Spatiotemporal_Map` function. The latter contains SpaceFlow's Leiden clustering (see appended Figure, *panel a*), and this result was used as the input for the microglia activation analysis. To further demonstrate that the source of the clustering information in the workflow does not affect outcomes, we also tested the opposite here, that is, by using the stLearn clusters in SpaceFlow (as opposed to its own). As anticipated, the calculated pseudotime values are similar for both workflows (see again appended Figure, *panel b,c*), and they are thus not a factor of influence in the outcome. Overall, we conclude that the previously presented analysis stand without change, and we trust that the additional clarification provided here has taken away the expressed concern and satisfied the reviewer.

MINOR CONCERNS:

5. There is some funny typesetting in the methods presumably due to underscores in num_exp_genes being typeset incorrectly. Amended and apologies for these oversights. They have been corrected in the revised version of the manuscript.

6. There are several broken references (Fig. ??, etc.) in the manuscript. We thank the reviewer for altering us to this. These have now also been corrected in the revised version of our manuscript.

APPENDIX 1

Figure: Comparison of pseudotime values with different clustering workflows for SpaceFlow. **a.** Original workflow used for benchmarking against SpaceFlow (described in main paper). **b.** SpaceFlow clusters and calculated pseudotime values (pSM). **c.** SpaceFlow pseudotime values (*right*) when calculated with stLearn clustering results (*left*). Note that the clustering input does not have an impact on pSM values.

Reviewer comments, third round

Reviewer #1 (Remarks to the Author):

The authors have fully addressed my comments.